# Application of Factor and Cluster Analyses to Determine Source–Receptor Relationships of Industrial Volatile Organic Odor Species in a Dual-Optical Sensing System

Jen-Chih Yang[1, 2], Pao-Erh Chang[1], Chi-Chang Ho[3], Chang-Fu Wu*[2]

[1]Green Energy and Environment Research Laboratories, Industrial Technology Research Institute, Room 220, 2F, Bldg.6, 321, Sec.2, Kuang Fu Rd., Hsinchu City 30011, Taiwan;

[2]Institute of Environmental and Occupational Health Sciences, National Taiwan University, Room 717, No. 17, Xu-Zhou Rd., Taipei 10055, Taiwan;

[3]Institute of Environmental and Occupational Health Sciences, National Taiwan University, Room 717, No. 17, Xu-Zhou Rd., Taipei 10055, Taiwan.

*Correspondence to: Chang-Fu Wu, Institute of Environmental and Occupational Health Sciences, National Taiwan University, Room 717, No. 17, Xu-Zhou Rd., Taipei 10055, Taiwan, R.O.C.; Tel./ fax: +886 2 3366 8096; E-mail: changfu@ntu.edu.tw

**Abstract.** Most odor nuisance investigations rely on either human olfactory examination or on-site sampling and analytical techniques, but these methods are often subject to spatial and temporal limitations and thus impractical for locating odor emission sources. This study developed an alternative approach with a dual-optical sensing system, a meteorological station, and the combination of factor and cluster analyses to identify and characterize emission sources of multiple air contaminants. Factor and cluster analyses were employed to establish the emission profile of multiple odorous substances from each emission source. Both receptor and source monitoring data were collected to characterize the emission sources of various odorous substances. Open-path Fourier transform infrared (OP-FTIR) as a receptor path detected concurrent trends of several organic solvents with concentrations higher than the reference odor threshold values, indicating that these compounds were potential causes of odor nuisance. Qualitative source apportionment by factor and cluster analyses suggested that these odorous substances were used as organic solvents in surface coating or painting processes. Closed-cell Fourier transform infrared (CC-FTIR) at two nearby surface-coating companies indicated that only one company's stack exhibited the same odorous substance profile found by the OP-FTIR receptor path. The major odor emission source was thus identified in this study. This study demonstrated the feasibility of using the alternative investigative framework to successfully identify emission sources from an industrial odor nuisance site. The major emission sources were identified, and future enforcement plans can be conducted to enhance odor investigation efficiency and improve overall air quality.

**Keywords.** Cluster analysis; dual-optical sensing system; qualitative source apportionment; factor analysis; Industrial odor emission sources; surface coating/painting

## 1. Introduction

The rapid growth of the economy and industrialization have led to environmental pollution problems, and consequently, an increase in environmental nuisance complaints has been evidenced in recent years. With more than 93,265 complaints, representing 33.7% of total reported environmental nuisances (Fig.1), odor nuisances have been ranked as the leading cause of environmental nuisances in Taiwan (Taiwan Environmental Protection Agency 2017). Volatile organic compounds (VOCs) are one of the factors contributing to odors and triggering various health problems, such as asthma, pneumonia, and bronchitis (Pride et al., 2015). It is also a precursor to fine particulate matter in the atmosphere, aggravating photochemical smog conditions in urban areas (Hu et al., 2017;Jathar et al., 2014). With residential area gradually expanded to industrial districts, odor nuisance has become another critical problem related to industrial VOC emissions, with a great impact on the quality of life.

Identifying the emission sources responsible for VOCs and odors remains a great challenge. Most odor nuisance investigations rely on either human olfactory examination or on-site sampling and analytical techniques (Merlen et al., 2017). However, these methods are hampered by spatial and temporal limitations. The "triangle odor bag method", originally developed by the Tokyo metropolitan government in 1972, was adopted by the Taiwanese government as a regulatory enforcement method in odor nuisance investigation. This method quantifies odor nuisance by using the human olfactory sense of a group of trained personnel (Higuchi, 2009;Higuchi and Masuda, 2004;Ueno et al., 2009). However, this method can only help determine the degree of odor intensity in collected air samples, but cannot enable the identification of the responsible emission sources. Sampling tools such as the Summa canister, Tedlar bag, and charcoal tube can be equipped with conventional fixed-point sampling and analytical methods to measure various VOC odor species (van Harreveld, 2003;Rumsey et al., 2012). However, these methods are highly temporally and spatially dependent, rendering the sampling of periodic or occasional odor episodes problematic. The insufficiency of conventional fixed-point sampling and analytical methods poses a great challenge to regulatory inspectors when odor nuisance occurs intermittently or during nonworking hours or originates from multiple sources. Many repeated air pollution complaints remain unresolved because the root pollution sources have not yet been found.

Fourier transform infrared (FTIR) spectrometry is an optical sensing technology that can detect multi-gaseous pollutants on a continuous basis and is, therefore, suitable for use in VOC or odor emission source investigation (Russwurm et al., 1991;Sung et al., 2014). It can allow real-time monitoring and analysis of several compounds simultaneously. The IR "fingerprints" of over 300 compounds were established on the basis of information from the US Environmental Protection Agency (USEPA) and the FTIR software developers. FTIR systems are of two types, namely open-path and closed-cell systems (USEPA, 2011). The open-path system, also called open-path Fourier transform infrared (OP-FTIR) spectroscopy, is an optical remote sensing technique used for measuring VOCs and inorganic compounds such as ammonia and hydrogen chloride in the ambient environment (e.g., fenceline

monitoring). The closed-cell system, also called closed-cell Fourier transform infrared (CC-FTIR) spectroscopy, is equipped with the same basic FTIR module as the OP-FTIR system, but employs gas pumps and sampling tubes to extract waste gas (e.g., from stack outlets) to a multipath cell attached to the FTIR spectrometer. In this study, the OP-FTIR and CC-FTIR systems were combined to obtain a "dual-optical sensing system" for accomplishing

the multiple functions of open-path long-range measurement, continuous monitoring, and multiple species measurement of stack exhaust, offering a powerful alternative method for investigating VOC or odor emission sources. Because OP-FTIR and CC-FTIR systems can generate a large speciation dataset in a short period, statistical methods play an essential role in data processing to extract the underlying meaning behind time-series patterns. Multivariate statistical modeling is suitable for processing FTIR data because it primarily analyzes

correlations between time-series trends of different species at different locations. By identifying common contaminants and concurrent trends among the various species measured using both systems, data from both receptors (OP-FTIR) and sources (CC-FTIR) may be compared and analyzed.

The aim of this study was to develop an alternative investigative framework to detect air pollution sources by using a dual-optical sensing system, a meteorological station, and factor and cluster analyses to enable future

accomplishments of emission reductions according to the investigation result.

## 2.  Materials and Methodology

### 2.1 Site description

Taiwan Environmental Protection Agency (TEPA) frequently receives complaints of odor nuisance at an intersection near an industrial park in southern Taiwan. The odor, being described as solvent- or chemical-like, is

20 mostly reported by commuters traveling through or waiting at the traffic signal at this intersection. A sunglass factory (hereafter called CY) is located at the northwest corner, a light metal casing factory (hereafter called KS) in the southeast corner, and a solar cell manufacturer (hereafter called NS) at a slight remove from the intersection to the east. Stacks (approximately 15–30 m high) on the rooftop of each factory continuously emit various processing gases during operating hours. The chemicals used at both CY and KS are mainly paint-related materials

containing organic solvents, such as toluene, xylene, acetone, and ethyl acetate, for surface coating purposes (CRC, 2006). NS mainly uses inorganic materials such as ammonia, silane, and nitric acid for silicon glass processing, thus generating both primary and secondary air pollutants (e.g., nitrogen dioxide) from high-temperature glass sintering processes (USPatent:4883521A, 1989).

### 2.2 Sampling techniques

To investigate odor emission sources at this location, an OP-FTIR beam path was deployed at the intersection to mimic the olfactory sense of people traveling through it. The OP-FTIR spectrometer (AirSentry-FTIR, CEREX,

USA) used in this study was a monostatic type equipped with Zn-Se beam splitters and liquid nitrogen-cooled mercury cadmium telluride (HgCdTe) detectors and a corner-cube retroreflector (PLX, Inc., USA) placed on the other end of the beam path. An infrared (IR) light beam transmitted from a telescope to the retroreflector targeted some distance away from the light emitter was reflected back to the detector inside the instrument, enabling

measurement of pollutants transported through the light beam path.

Monitoring was conducted from March 9 to 19, 2015, collecting a total of 2,911 consecutive spectral data. The OP-FTIR beam path was 143 m long in one direction and was equipped with a light emitter on the ground level on one side and a retroreflector at a height of 10 m on the other side. A meteorological station at a height of 12 m (fourth floor) was used together with the OP-FTIR beam path to monitor wind speed and direction (see Fig. 2a).

Wind and OP-FTIR data were simultaneously measured and continuously collected in a synchronic system to enable identification of the incoming direction of gaseous contaminants and provide the spatiotemporal measurement of VOCs or odor pollutants.

Official documents were reviewed to ascertain the raw material usage of each of the nearby factories. Three potential sources (factories), namely, CY, KS, and NS were targeted for further stack monitoring using CC-FTIR. A

10-m (path length) gas cell with the inner pressure of 720 mm Hg, and an estimated gas flow rate of 0.37 Liter/sec. was used for the CC-FTIR multi-reflection gas measurements. The water vapor was mostly removed by using an impinger connected to the inlet of the gas cell to decrease interference with $H_2O$ absorption in the FTIR spectra. The stack exhaust of these three factories was measured for 24 to 242 hours, generating data at each 5-min interval. This continuous monitoring system generated sufficient time-series data to enable factor and cluster

analysis in the next phase. Two CC-FTIR systems were deployed at each selected emission source to measure chemical species of exhaust gases from each stack (see Fig. 2b). Sampling tubes were divided into several manifolds at the stack end, joining together before entering the CC-FTIR gas cell. This sampling method allowed multiple waste gas flow from different stacks to be collected and transferred to the gas cell simultaneously, avoiding time lags when switching the sampling line from one stack to another. A total of 4,378 spectral data was

collected from the stack outlets of the three potential odor emission sources, namely 288 spectra from CY, 2,907 spectra from KS, and 1,183 spectra from NS.

## 2.3 Chemical analysis methods

Any gaseous compounds absorbed in the IR region (approximately 2.5–25 microns) were potential candidates for monitoring using FTIR technology. The resolution of the OP-FTIR and CC-FTIR interferograms was 1 cm$^{-1}$,

recording a coadded infrared spectrum at 5-min intervals, with 64 IR scans generated at each interval. Contaminants of interest were identified and quantified using spectral search software featuring compound-specific analysis and comparison to the system's internal reference spectra library. The unique fingerprint characteristics of each chemical compound brought identification of gaseous pollutants possible

through comparing the shape, position and relative peak height of each measured spectrum with reference spectra. Multicomponent classical least-squares techniques were employed in the FTIR spectral quantitative analysis. Rolling backgrounds were used in OP-FTIR spectral analysis to eliminate baseline shifts resulting from contingent changes in weather conditions (Hunt, 1995). The rolling background was collected using the first

spectrum as a background to create an absorbance spectrum from the second spectrum, using the second spectrum as a background for the third spectrum and so on. The integral values of concentrations are calculated to obtain time-series data for each compound. The advantage of using the rolling background is that it will have the best correction for water vapor, detector and instrument response, and the lowest residual error (Hunt, 1995). A "fixed" reference method was used in CC-FTIR spectral analysis. The fixed reference method uses a reference

spectrum that is taken from the zero air or highly purified nitrogen to generate a bundle of spectra using an identical reference spectrum. The main advantage of this method is that the reference is pure, without any contaminants, and the absolute concentrations of the contaminants can be calculated accordingly (Hunt, 1995).

## 2.4 Qualitative receptor modeling

Factor analysis and cluster analysis using the SAS statistical software package (SAS Institute, Inc., USA) were

15 employed in qualitative receptor modeling in this study. Factor analysis uses an eigenvector with varimax orthogonal rotations to interpret large datasets (Johnson, 1998). The factor analysis model expresses each variable as a linear combination of underlying *common factors* $f_1$, $f_2$,…,$f_m$ with an accompanying error term to specify that part of the variables that are uncorrelated with any of the common factors. For $X_1$, $X_2$,…, $X_p$ in any observation vector **X**, the m-factor model is calculated using the following Eq. (1–4): (Rencher, 2002):

$$X_1 = a_{11}f_1 + a_{12}f_2 + … + a_{1m}f_m + e_1 \quad\quad \text{Eq. (1)}$$
$$X_2 = a_{21}f_1 + a_{22}f_2 + … + a_{2m}f_m + e_2 \quad\quad \text{Eq. (2)}$$
$$X_p = a_{p1}f_1 + a_{p2}f_2 + … + a_{pm}f_m + e_p \quad\quad \text{Eq. (3)}$$
$$\mathbf{X} = (X_1,…,X_P)', \; \boldsymbol{f} = (f_1,…f_m)', \text{ and } \mathbf{e} = (e_1,…e_p)' \quad\quad \text{Eq. (4)}$$

where $X_i$ = the i*th* chemical species with mean 0 and unit variance, i = 1,…,p; $a_{i1}$ to $a_{im}$ = the factor loadings for the i*th* chemical species; $f_1$ to $f_m$ = **m** uncorrelated common factors, each with mean 0 and unit variance; e = the error terms indicating the residual part of $X_i$ that is not in common with the other variables.

Because data collected by FTIR contain many intercorrelated variables that are multivariate, the simultaneous consideration of all variables was essential to understanding the underlying meaning of the measured data.

Variables (VOC or odor substances) with concurrent patterns were grouped together as a factor to gain insight into the underlying emission source characteristics. Factors with an eigenvalue greater than one were retained for varimax rotations and factor loading calculations. Factor loadings with absolute values greater than 0.4 were considered influential variables (Rencher, 2002); the higher the factor loading (>0.4), the stronger the correlation between the variables (odor substances) and the factor (emission source). The combination of variables in each

factor roughly represented the types or characteristics of each factor or source. This method is especially useful when the patterns of association between the receptor (measured by ambient OP-FTIR) and source (measured by stack CC-FTIR) are compared reciprocally, enabling emission sources that mutually correspond to be identified.

Cluster analysis is used to find patterns in a dataset by grouping all variables into clusters. A single linkage method (also called nearest neighbor method), a type of hierarchical methods, was used to calculate the distance between two clusters in this study. In the single linkage method, the distance between two clusters $A$ and $B$ is defined as the minimum distance between a point in $A$ and a point in $B$ described as Eq.(5) (Rencher, 2002):

$$D(A, B) = min\{d(y_i, y_j), for \ y_i \ in \ A \ and \ y_j \ in \ B\} \qquad \text{Eq. (5)}$$

where $d(y_i, y_j)$ is the Euclidean distance (Rencher, 2002, chapter 14)

The concurrent trends between different species can be analyzed using both factor and cluster analysis. Odor contaminants with concurrent patterns were grouped as a factor to gain insight into the underlying emission source characteristics. Meteorological data was used to confirm the factor analysis in the way that the incoming wind direction of each factor (representing a group of chemicals) may be different according to the relative locations of each potential odor sources. Cluster dendrograms provide linkage paths between groups of chemicals to offer more information about the characteristics of different emission sources.

## 3. Results and Discussion

### 3.1 Meteorological data

The meteorological data from March 9 to 19, 2015 are shown in Fig. 3. The prevailing wind from March 9 to 14 was from the NNW–N–NNE direction, whereas the prevailing wind from March 15 to 18 was from the SSW–S–SSE–SE–ESE direction. A dramatic change in wind direction from March 14 to March 15, when the incoming wind direction changed from north to south, was observed. The integrated wind direction is shown in Fig. 3a indicated that the overall wind direction was from the N–NNE direction during the 10 days of field monitoring.

### 3.2 Ambient data from receptor path

Table 1 shows the ambient concentration of air contaminants measured using the OP-FTIR system at the intersection. The first column represents the 16 species measured by the receptor path (OP-FTIR), namely acetone, ethyl acetate, ammonia, gasoline, m-xylene, nitrogen dioxide, o-xylene, n-butyl acetate, toluene, propylene glycol monoethyl acetate (PGMEA), p-xylene, acetylene, ethylene, butyl cellosolve, carbon monoxide, and nitrous oxide. Fig. 4 displays a series of comparisons between the measured and reference spectra. The concentrations of most species were quantified, except for background species such as carbon monoxide and nitrous oxide. The exact concentration of background species cannot be quantified using a rolling background in

the spectral analysis because of unknown background levels; however, the incremental concentration of these species can still be calculated to generate concentration trends suitable for factor analysis. A total of 2,911 consecutive spectra was collected during the 10 days of field monitoring, with various detection limits intrinsic to each compound. The numbers shown in the second column indicated that the probability of detection of ammonia,

ethyl acetate, acetone, butyl cellosolve, n-butyl acetate, o-xylene, PGMEA, and ethylene was higher than that of other species. The maximum value of each detected contaminant represented the highest concentration measured within a 5-min period. Concentrations detected using OP-FTIR were path average. Among the 16 detected species, the major compounds were gasoline, m-xylene, and nitrogen dioxide, with mean concentrations of $33.21 \pm 5.00$, $27.96 \pm 6.05$, and $25.13 \pm 3.28$ ppbv, respectively. Toluene, o-xylene, and acetone revealed mean

concentrations ranging from 11.61 to 20.57 ppbv. The concentration levels of gasoline, m-xylene, nitrogen dioxide, n-butyl acetate, toluene, and PGMEA were higher than the odor threshold reference values, indicating that these compounds were potential causes of odor nuisance in the intersection zone. These odor substances are mainly used as organic solvents in surface coating or painting processes. The evidence of the correlation between the substances (concentrations) detected at the receptor site and reported odor nuisance events was provided by

using phi coefficients and point biserial correlation (Gallagher, 2011;Demirtas et al., 2012). The phi-coefficient correlations ($r_{phi}$) for "odor" versus "compound" displayed correlation coefficients of two dichotomous variables between the detection of compounds (detected vs. non-detected) and the perception of odor (odor vs. non-odor; as recorded by the local environmental protection agency). The point biserial correlation ($r_{pb}$), a correlation between one continuous and one dichotomous variable, represents the concentration of compounds and the

perception of odor (Capelli et al., 2013). A value close to 1 for $r_{phi}$ / $r_{pb}$ indicated that the association between "odor" and "compound" was strong. The $r_{phi}$ / $r_{pb}$ between the "odor" and acetone, ethyl acetate, toluene, PGMEA and butyl cellosolve were mostly at moderate levels ($r_{phi}$ = 0.50 to 0.67; $r_{pb}$ = 0.30 to 0.45), and the correlations were statistically significant ($p<0.001$). Relatively weak correlations between the "odor" and m-xylene, p-xylene, and n-butyl acetate were shown, although the correlations were statistically significant ($p<0.001$) as well. Therefore, it

would suggest that acetone, ethyl acetate, toluene, PGMEA, and butyl cellosolve were the most possible odor substances that were correlated with the recorded odor nuisance events, which were defined by any solvent smell arising from the intersection zone. A complete time-series pattern of chemical species found at the receptor site that was used as the basis for the calculation of r_phi and r_pb was shown in Fig. 5, in which the periods when the odor was reported were highlighted.

Table 2 summarizes the results of factor analysis for the OP-FTIR receptor path. The pattern of the first factor (F1_OP) indicated several organic solvents, including m-xylene, p-xylene, o-xylene, ethyl acetate, PGMEA, toluene, and butyl cellosolve, all of which are commonly used as chemical solvents in surface coatings and paints (USEPA, 2009) and could be considered possible causes of odor nuisance because their concentrations were higher than the reference values. The daytime pattern of factor scores for the first group, as shown in Fig. 6a,

revealed higher concentrations and frequencies of occurrence from 14:00 to 22:00, particularly on weekdays. This could explain the higher incidence of odor nuisance complaints during the afternoon and evening hours on weekdays. Moreover, the incoming direction of these seven species (as represented by factor scores) revealed that the highest factor score occurred in the direction of the WNW, although a few came from the direction of ESE

and the directions of NNW–ENE (Fig. 7a).

The compounds included in the second factor (F2_OP) were acetylene, ethylene, gasoline, and carbon monoxide. Fig. 6b shows the daytime pattern of these five species, indicating higher concentrations during the peak traffic hours from 6:00 to 9:00 and 17:00 to 20:00 on weekdays (Fig. 6b). This unique pattern indicates that the second group of compounds may be derived from incomplete combustion of vehicles waiting or idling at the intersection

and thus generating chemical byproducts such as acetylene, ethylene, and carbon monoxide (USEPA, 2000;Liu et al., 2014). The incoming directions of Factor 2 were mostly from NNE–NE, although a few came from the directions of ENE–SE and the direction of NNW (Fig. 7b), indicating multiple source directions for the incomplete engine combustion.

Ammonia, nitrogen dioxide, and nitrous oxide were identified as the third-factor (F3_OP) compounds. These

mainly inorganic compounds exhibited higher concentrations from 06:00 to 09:00 on weekends (Fig. 6c) mostly came from the  NNE–ESE directions, although a few came from the SSW direction (Fig. 7c), indicating that the major upwind location of the emission source(s) was located in the NNE–ESE direction. The solar cell production company located in the East direction and using inorganic materials such as ammonia, silane, and nitric acid to produce silicon glass could generate nitrogen dioxide and nitrous oxide from high-temperature glass sintering

processes (USPatent:4883521A, 1989), and was therefore deemed the potential emission source.

The fourth-factor (F4_OP) compounds, namely acetone and n-butyl acetate, also exhibited higher concentrations and greater frequency of occurrence from 6:00 to 10:00 and from 17:00 to 22:00 on weekdays (Fig. 6d). The incoming direction of these two compounds was mainly from the N–ENE direction (Fig. 7d), which is slightly different from that of the first-factor (F1_OP) compounds.

The four factors were identified and characterized through the combination of species, hours of emission, and incoming direction of each. Four groups of emission sources were identified and categorized using factor analysis, namely surface coating (paint), incomplete engine combustion, solar cell production, and solvent use.

**3.3 Comparison of ambient data from the receptor path and source profiles from multiple stacks**

The ambient data from the receptor path indicated several factors or source groups at the intersection, including

organic solvents from the surface coating, traffic emissions from incomplete vehicle engine combustion, and inorganic emissions from solar cell production. Official documents showed that the chemicals used in both CY and KS were paint-related materials containing organic solvents, which were thus categorized as first-factor (F1_OP) compounds. However, windrose diagrams for the first factor (Fig. 7a) displayed multiple source directions

(including N–NE, NW, and ESE), indicating that the first factor (F1_OP) might not be limited to one source; further efforts are thus required to clarify the sources. To analyze observations at the receptor path, the emission profiles of potential sources were compared.

Figure 5 and Table 3 present a comparison of the detected air pollutants and their concentrations at the receptor path and source stacks in the intersection zone. Vehicle exhaust profiles from the USEPA's SPECIATE database are also provided in the last column of Table 3 to indicate the emissions from traffic incomplete vehicle engine combustion at the receptor path. Almost every compound detected in the receptor path corresponded with one or more chemicals from the source stacks, except for traffic-related chemicals (e.g., gasoline, ethylene, and acetylene). The panel plot shows the patterns of association between the receptor (ambient data from OP-FTIR) and the source (stack source profile from CC-FTIR). Concentration boxplots for chemical species (except carbon monoxide) measured using OP-FTIR (at the intersection) are shown in Fig. 8e, with eight species coinciding with those found in the CY stacks (Fig. 8a), seven coinciding with those found in the KS stacks (Fig. 8b), and three coinciding with those found in the NS stacks (Fig. 8c), as well as six from vehicle emissions (Fig. 8d). Furthermore, among the species found in the CY and KS stacks, six coexisted in both factories, namely, ethyl acetate, toluene, o-xylene, m-xylene, p-xylene, and acetone, indicating that these six compounds were common species emitted at both locations. By contrast, butyl cellosolve and PGMEA were uniquely found in the CY stacks. Ammonia was found at both the KS and NS stacks.

### 3.4 Factor and cluster analyses of sources

Because the chemicals used at both CY and KS were mainly organic solvents that are similar to each other, factor analysis was performed for each source to distinguish the main contributor of odor nuisance in this location and examine relationships between the ambient data and the profiles of these two sources.

Two types of multivariate statistical methods, namely factor and cluster analyses, were used together to analyze concurrent trends of CC-FTIR data measured at the CY, KS and NS stacks (Table 4). The result of factor analysis for CY (Table 4a) indicated two factors with an eigenvalue greater than one. The influential species (factor loading of >0.4) for the first factor (F1_CY) were o-xylene, m-xylene, p-xylene, toluene, PGMEA, ethyl acetate, and butyl cellosolve, but only acetone for the second factor (F2_CY). The first factor (F1_CY) contained a combination of various types of solvents used as paint thinners (for plastic coating purposes), whereas the second factor (F2_CY) species (acetone) were used as a chemical solvent to remove residual paint in sprinkle nozzles. Two factors were also identified from the CC-FTIR results for the KS stack (Table 4c). The first factor (F1_KS) comprised p-xylene, toluene, m-xylene, and o-xylene, and the second factor acetone and ethyl acetate. The first factor (F1_KS) thus contained various chemical solvents used as paint thinners (for metal coating purposes), whereas the second factor (F2_KS) contained substances used for cleaning or other purposes in manufacturing light metal casings. The chemicals from the NS stacks were mainly inorganic materials (nitrous oxide, silane, ammonia, nitrous acid,

and nitrogen dioxide) that were either primary or secondary air pollutants derived from solar cell production (Table 4e), all of which did not correspond with the organic odorous solvents identified in the receptor sites. The first factor (F1_NS) was comprised of raw materials used for growing anti-reflection films, including nitrous oxide, silane, and ammonia, in which silane and ammonia were often controlled in opposite flow rates to ensure no

significant pressure fluctuations (ChinaPatent:CN102244109B, 2013). The second factor (F2_NS) contained nitrous acid and nitrogen dioxide, in which the formation of $NO_2$ is enhanced by thermal decomposition of $HNO_3$.

Using cluster dendrograms, different compounds can be linked to represent their relationships with each other and the interrelationships between groups, thus providing another means of displaying correlations between different variables. According to the cluster analysis results in Table 4b& 4d, acetone was excluded from other chemicals

already in the first branch, indicating that its source was different from others. Similarly, the linkage path between groups of chemicals differed from one company to another, indicating that different types of paint thinner could be used in two companies for different purposes. Factor analysis between ambient data and source profiles indicated that the grouping pattern of seven odorous compounds (o-xylene, m-xylene, p-xylene, toluene, PGMEA, ethyl acetate, and butyl cellosolve) between the receptor path (OP-FTIR) and the CY stack (CC-FTIR) was identical.

Thus, the CC-FTIR results from the CY stacks indicated the same odorous compounds as the receptor path (OP-FTIR), all of which came from the direction of CY. However, the grouping pattern for KS differed from that of the receptor path (OP-FTIR), with three key species in the first factor (PGMEA, ethyl acetate, and butyl cellosolve) missing in the KS stacks.

Figure 9 uses scatter plots to display concentration variations in selected contaminants over time, with the

interrelationship between odorous compounds at the CY stack (CC-FTIR) and the receptor path (OP-FTIR) delineated and compared. Compounds for each pair were linearly correlated, with the correlation coefficients mostly greater than 0.7. However, the correlation coefficients for the KS stack was mostly below 0.1, indicating that the relationships between the ambient data and the KS source profiles were not as significant as those for CY.

## 4. Conclusion

This study developed an alternative investigative framework for detecting air pollution sources of odor nuisance by measuring 16 gas species simultaneously using FTIR spectroscopic measurements and factor analyses to identify and characterize emission sources of multiple air contaminants. Meteorological data and cluster analysis were employed to proof the identification of the major odor emissions. Different industrial processes were related to a specific combination of different pollutants, and this combination was obtained using the two statistical methods of

factor analysis and cluster analyses. Factor and cluster analyses were employed to improve the quality and completeness of the source profiles. A field study used FTIR spectroscopic measurements to determine the source of the emission of volatile organic odor species near an industrial park in southern Taiwan demonstrated

the feasibility of this proposed method. The major odor emission source was identified through qualitative source apportionment of factor and cluster analyses. With enhanced efficiency in odor investigation methodology, future emission reduction plans can be developed and overall air quality can be improved.

## 5. Data availability

The data that support the findings of this study are not publicly available due to the containing information could compromise the privacy of research participants.

## 6. Author contribution

Jen-Chih Yang and Pao-Erh Chang designed the experiments. Jen-Chih Yang performed the spectra analysis of both OP-FTIR and CC-FTIR systems. Jen-Chih Yang and Chi-Chang Ho performed statistical modeling for factor
and cluster analyses. Jen-Chih Yang prepared the manuscript with contributions from all co-authors. Chang-Fu Wu supervised the project.

## 7. Competing interests

The authors declare that they have no conflict of interest.

## 8. Acknowledgments

The authors wish to express their gratitude for the administrative support from both Taiwan Environmental Protection Agency and Tainan City Environmental Protection Bureau.

## 9. Financial support

This open-access publication was funded by Green Energy and Environment Research Laboratories, Industrial Technology Research Institute.

## 10. Review statement

This manuscript was edited by Wallace Academic Editing and reviewed by two anonymous referees.

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

**Table 1: Descriptive statistics of VOC measurements at the receptor site and the correlation coefficients between the receptor site and the reported odor nuisance events**

| Compound | Detected n | Concentration (ppbv) Max[b] | Concentration (ppbv) Mean±SE | CV (%) | Odor Threshold (ppbv) | MDC[c] (ppbv) | Odor vs. Compound Correlation $r_{phi}$[d] | Odor vs. Compound Correlation $r_{pb}$[e] | p-value[f] $r_{phi}$[d] | p-value[f] $r_{pb}$[e] |
|---|---|---|---|---|---|---|---|---|---|---|
| Acetone | 182 | 130.3 | 11.61±1.02 | 118.8 | 37 | 12.3 | 0.672 | 0.432 | <0.001*** | <0.001*** |
| Ethyl Acetate | 519 | 126.1 | 4.95±0.38 | 176.5 | 170 | 4.2 | 0.551 | 0.450 | <0.001*** | <0.001*** |
| Ammonia | 982 | 117.6 | 7.07±0.50 | 223.0 | 45 | 2.7 | 0.286 | -0.129 | <0.001*** | 0.046* |
| Gasoline (a mixture of C5+ and BTEX) | 22 | 110.2 | 33.21±5.00 | 70.6 | 25 | 11.5 | 0.155 | 0.016 | 0.017* | 0.810 |
| m-Xylene | 17 | 106.4 | 27.96±6.05 | 89.2 | 81 | 12.5 | 0.328 | 0.272 | <0.001*** | <0.001*** |
| Nitrogen Dioxide | 35 | 98.1 | 25.13±3.28 | 77.1 | 58 | 19.6 | -0.161 | -0.107 | 0.013* | 0.099 |
| o-Xylene | 96 | 72.0 | 16.26±0.89 | 53.8 | 180 | 8.8 | -0.001 | 0.027 | 0.987 | 0.680 |
| n-Butyl Acetate | 103 | 57.7 | 9.92±1.05 | 107.2 | 6.3 | 4.6 | 0.345 | 0.020 | <0.001*** | 0.758 |
| Toluene | 16 | 54.5 | 20.57±3.36 | 65.3 | 21 | 18.0 | 0.328 | 0.302 | <0.001*** | <0.001*** |
| Propylene Glycol Methyl Ether Acetate (PGMEA) | 93 | 46.3 | 4.62±0.67 | 139.8 | 25 | 1.8 | 0.569 | 0.312 | <0.001*** | <0.001*** |
| p-Xylene | 39 | 43.6 | 8.15±1.13 | 87.0 | 120 | 5.7 | 0.168 | 0.215 | 0.010* | 0.001** |
| Acetylene | 65 | 17.4 | 3.80±0.35 | 75.1 | 226000 | 1.7 | 0.016 | -0.017 | 0.803 | 0.791 |
| Ethylene | 71 | 12.0 | 3.17±0.25 | 67.0 | 17000 | 1.9 | 0.166 | 0.063 | 0.010* | 0.331 |
| Butyl cellosolve | 137 | 18.5 | 2.30±0.20 | 104.2 | 100 | 2.6 | 0.499 | 0.304 | <0.001*** | <0.001*** |
| Carbon monoxide[a] | 2779 | | | | | Detected | | | | |
| Nitrous oxide[a] | 15 | | | | | Detected | | | | |

Measurements performed from March 9, 2015 at 15:41 to March 19, 2015 at 11:25; 2,911 spectra were recorded at 5-min intervals. [a] Background species. The exact concentration of the background species cannot be quantified using a rolling background because of unknown background levels. [b] Underlined numbers represent concentrations exceeding corresponding odor thresholds. [c] MDC (estimated minimum detectable concentrations) is calculated based on path length 286m (two-way), 5 min average by the peak-to-peak (p-p) absorbance noise in the spectral region of the target absorption feature and the MDC is the absorbance signal (of the target compound) that is equal to the p-p noise level, using a reference spectrum acquired for a known concentration of the target compound:

$$MDC = \frac{(ppm * m)}{A_n(v)} * \frac{NEAx}{pathlength(m)}$$

where $MDC_{peak-to-peak}$= estimated minimum detectable concentration (ppm or ppb), An(v) = normalized absorbance, $NEA_x$= noise equivalent absorbance is peak-to-peak noise, and path length = two way path length (m)

[d] Phi Correlation Coefficient (Gallagher, 2011). [e] Point Biserial Correlation (Demirtas et al., 2012). [f] Statistically significant correlation coefficients are marked with *p < 0.05, **p<0.01, *** p<.0.001

**Table 2:** The grouping of the data as a function of time and wind direction using factor analysis for chemical species measured by OP-FTIR at the receptor site

| Compounds | MSA[b] | Factor #_OP (OP-FTIR)[a] (variance explained) | | | |
|---|---|---|---|---|---|
| | | F1_OP (28.15%) | F2_OP (13.13%) | F3_OP (7.42%) | F4_OP (6.18%) |
| m-Xylene | 0.84 | ***0.95***[c] | 0.02 | 0.00 | 0.01 |
| PGMEA | 0.83 | ***0.95***[c] | 0.04 | 0.00 | 0.18 |
| Ethyl Acetate | 0.93 | ***0.90***[c] | 0.06 | -0.02 | 0.24 |
| p-Xylene | 0.93 | ***0.89***[c] | 0.03 | 0.00 | 0.03 |
| Toluene | 0.94 | ***0.85***[c] | -0.02 | 0.01 | 0.04 |
| Butyl cellosolve | 0.85 | ***0.78***[c] | 0.02 | 0.01 | 0.30 |
| o-Xylene | 0.97 | ***0.63***[c] | -0.01 | -0.03 | -0.16 |
| Acetylene | 0.66 | 0.02 | ***0.88***[c] | -0.03 | -0.02 |
| Ethylene | 0.68 | 0.04 | ***0.86***[c] | -0.04 | -0.03 |
| Gasoline | 0.81 | -0.01 | ***0.69***[c] | 0.01 | -0.03 |
| Carbon Monoxide | 0.83 | 0.03 | ***0.60***[c] | 0.19 | 0.17 |
| Nitrogen Dioxide | 0.52 | 0.00 | 0.01 | ***0.82***[c] | -0.04 |
| Ammonia | 0.57 | -0.02 | 0.25 | ***0.81***[c] | -0.04 |
| Nitrous Oxide | 0.50 | 0.00 | -0.05 | ***0.37***[c] | 0.03 |
| Acetone | 0.65 | 0.01 | 0.00 | 0.00 | ***0.76***[c] |
| n-Butyl Acetate | 0.81 | 0.19 | 0.05 | 0.00 | ***0.63***[c] |
| *Possible sources* | | *Surface Coating (paint thinner)* | *incomplete engine combustion* | *Solar cell production* | *Solvent use (paint remover)* |
| *Major source directions* | | *WNW, ESE, NNW–ENE* | *NNE-NE, ENE–SE, NNW* | *NNE–ESE, SSW* | *N–ENE* |

[a] Extraction method: principal component analysis; rotation method: varimax with Kaiser normalization . [b] Kaiser's measure of sampling adequacy: overall MSA = 0.849, indicating the dataset's appropriateness for use in factor analysis is meritorious.; [c] bold underlined numbers represent factor loadings of >0.40, indicating the main species in each factor (source);.

**Table 3: Concentrations of the receptor path versus source stacks and vehicle exhaust profile (in ppbv)**

| Compounds | Max. Receptor (OP-FTIR) Intersection (*n*=2,911) | Mean Sources (Stack Emissions) (CC-FTIR) Plant CY (*n*=288) | Plant KS (*n*=2,907) | Plant NS (*n*=1,183) | Speciate (Traffic Profile) Vehicle Exhaust |
|---|---|---|---|---|---|
| Ethyl Acetate | 126.1 | 8,045.2 | 228.4 | --. | -- |
| Toluene | 54.5 | 11,085.4 | 12.0 | --. | 57,200 |
| o-Xylene | 72.0 | 7,063.4 | 12.9 | --. | 17,600 |
| m-Xylene | 106.4 | 5,102.2 | 9.4 | --. | -- |
| p-Xylene | 43.6 | 4,377.3 | 3.6 | --. | 28,700 |
| Acetone | 130.3 | 15,396.5 | 15,151.1 | --. | -- |
| Butyl Cellosolve | 3.1 | 2,413.1 | --. | --. | -- |
| PGMEA | 46.3 | 949.1 | --. | --. | -- |
| Ammonia | 117.6 | -- | 350.0 | 9,723.1 | -- |
| Nitrous Oxide | detected | -- | --. | 2,040.7 | -- |
| Nitrogen Dioxide | 98.1 | -- | --. | 234.9 | -- |
| Nitrous acid | -- | -- | --. | 1.1 | -- |
| Silane | -- | -- | --. | 31.9 | -- |
| n-Butyl Acetate | 57.7 | -- | 9.9 | --. | -- |
| Acetylene | 17.4 | -- | -- | -- | 22,500 |
| Ethylene | 12.0 | -- | -- | -- | 84,100 |

**Table 4: Factor & cluster analyses of chemicals measured using CC-FTIR in the stacks of 3 companies**

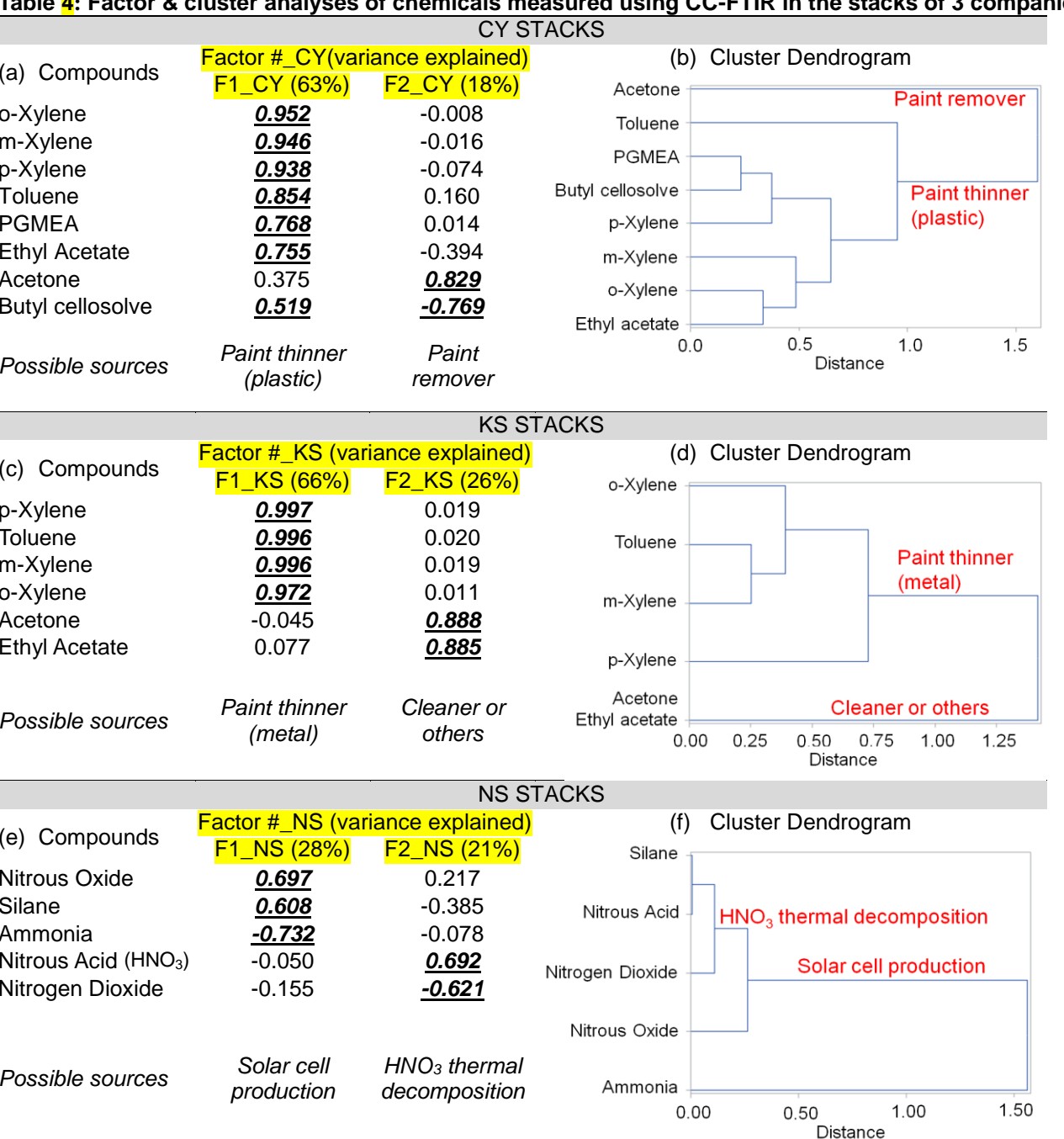

| CY STACKS | | | |
|---|---|---|---|
| (a) Compounds | Factor #_CY(variance explained) | | (b) Cluster Dendrogram |
| | F1_CY (63%) | F2_CY (18%) | |
| o-Xylene | *0.952* | -0.008 | |
| m-Xylene | *0.946* | -0.016 | |
| p-Xylene | *0.938* | -0.074 | |
| Toluene | *0.854* | 0.160 | |
| PGMEA | *0.768* | 0.014 | |
| Ethyl Acetate | *0.755* | -0.394 | |
| Acetone | 0.375 | *0.829* | |
| Butyl cellosolve | *0.519* | *-0.769* | |
| Possible sources | *Paint thinner (plastic)* | *Paint remover* | |

| KS STACKS | | | |
|---|---|---|---|
| (c) Compounds | Factor #_KS (variance explained) | | (d) Cluster Dendrogram |
| | F1_KS (66%) | F2_KS (26%) | |
| p-Xylene | *0.997* | 0.019 | |
| Toluene | *0.996* | 0.020 | |
| m-Xylene | *0.996* | 0.019 | |
| o-Xylene | *0.972* | 0.011 | |
| Acetone | -0.045 | *0.888* | |
| Ethyl Acetate | 0.077 | *0.885* | |
| Possible sources | *Paint thinner (metal)* | *Cleaner or others* | |

| NS STACKS | | | |
|---|---|---|---|
| (e) Compounds | Factor #_NS (variance explained) | | (f) Cluster Dendrogram |
| | F1_NS (28%) | F2_NS (21%) | |
| Nitrous Oxide | *0.697* | 0.217 | |
| Silane | *0.608* | -0.385 | |
| Ammonia | *-0.732* | -0.078 | |
| Nitrous Acid ($HNO_3$) | -0.050 | *0.692* | |
| Nitrogen Dioxide | -0.155 | *-0.621* | |
| Possible sources | *Solar cell production* | *$HNO_3$ thermal decomposition* | |

**13. Figures**

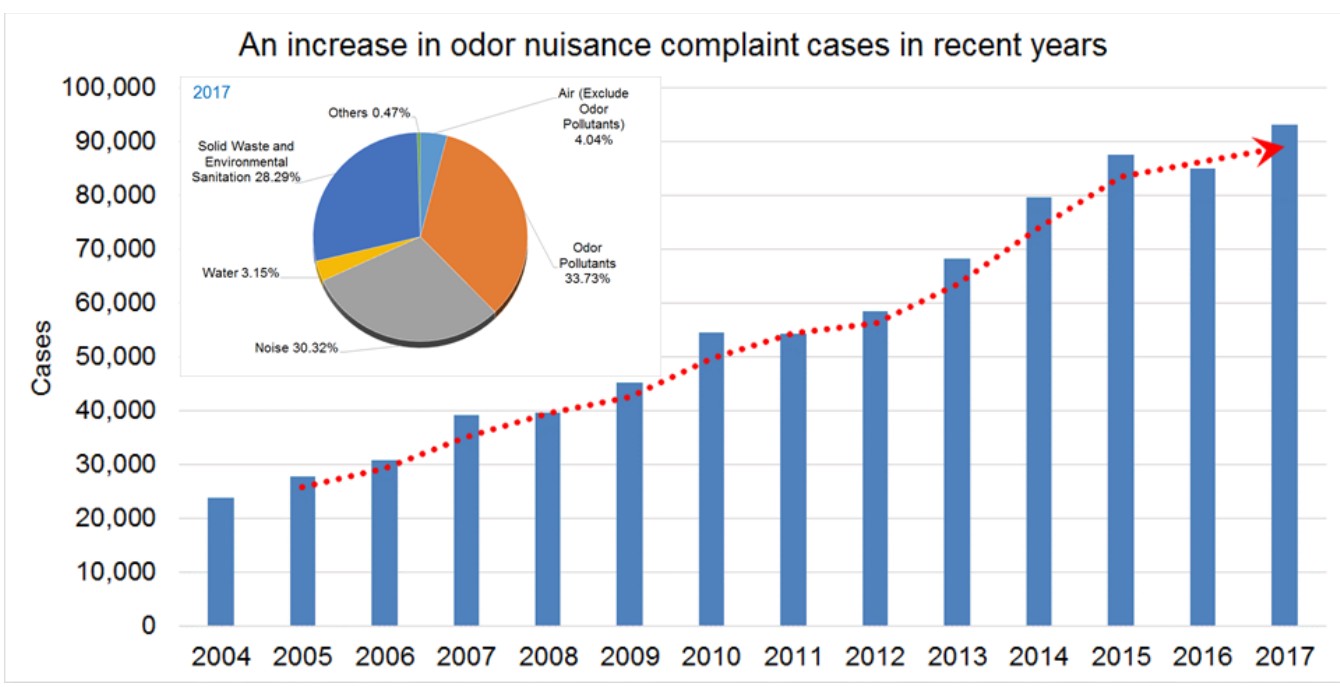

**Figure 1: Trend of total odor nuisance complaints by the TEPA from 2004 to 2017.** An increase in odor nuisance complaints has been evidenced in recent years and the odor nuisances have been ranked as the leading cause of environmental nuisances in Taiwan.

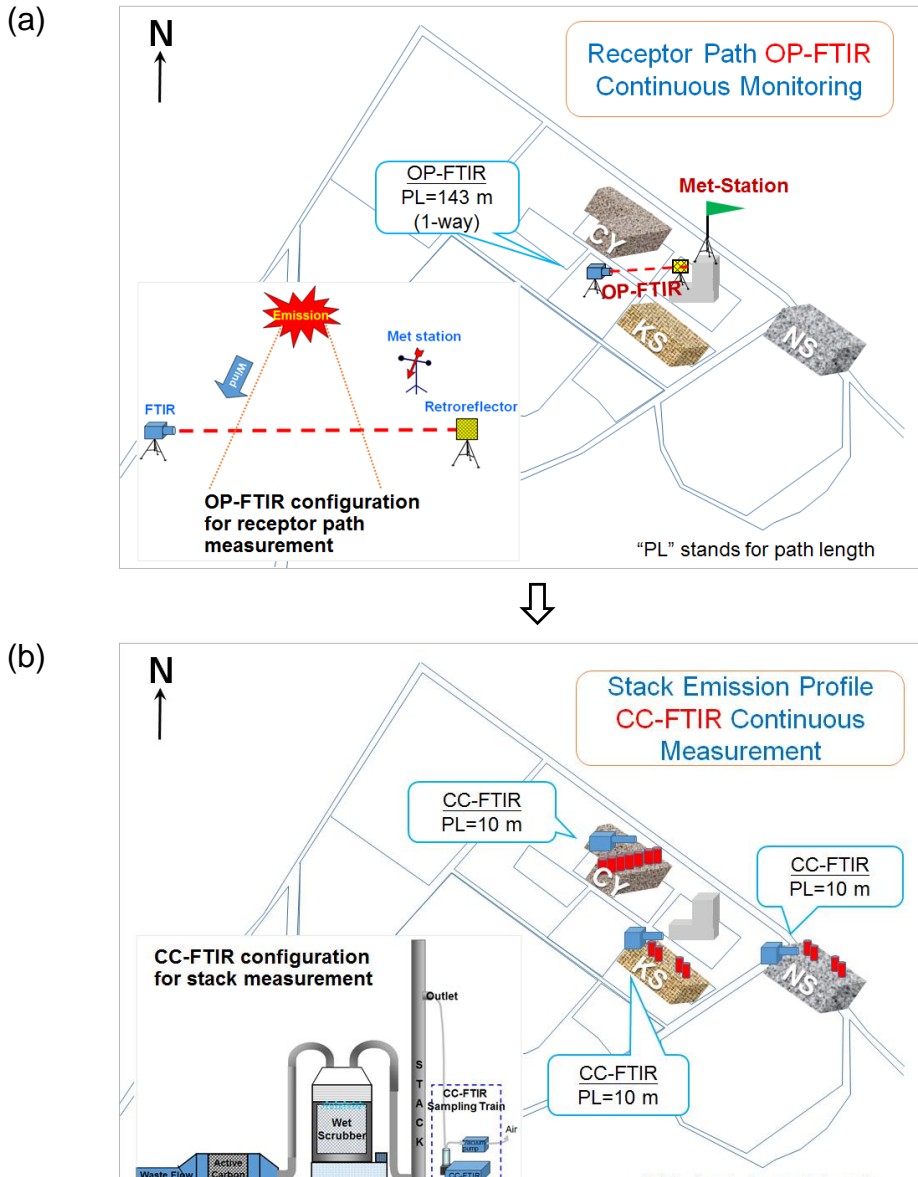

**Figure 2: Top view of OP- and CC-FTIR configuration; (a) Receptor path of OP-FTIR monitoring at the intersection. The OP-FTIR beam path was 143 m long in one direction and was equipped with a light emitter on the ground level on one side and a retroreflector at a height of 10 m on the other side. A meteorological station at a height of 12 m was used together with the OP-FTIR beam path to monitor wind speed and direction. Wind and OP-FTIR data were measured as a synchronic system to enable identification of the incoming direction of gaseous contaminants and provide a spatiotemporal measurement of VOCs or odor pollutants; (b) Source stack CC-FTIR measurement at three potential odor emission sources. A 10-m (path length) gas cell with the inner pressure of 720 mm Hg, and an estimated gas flow rate of 0.37 L/sec was used for the CC-FTIR multi-reflection gas measurements.**

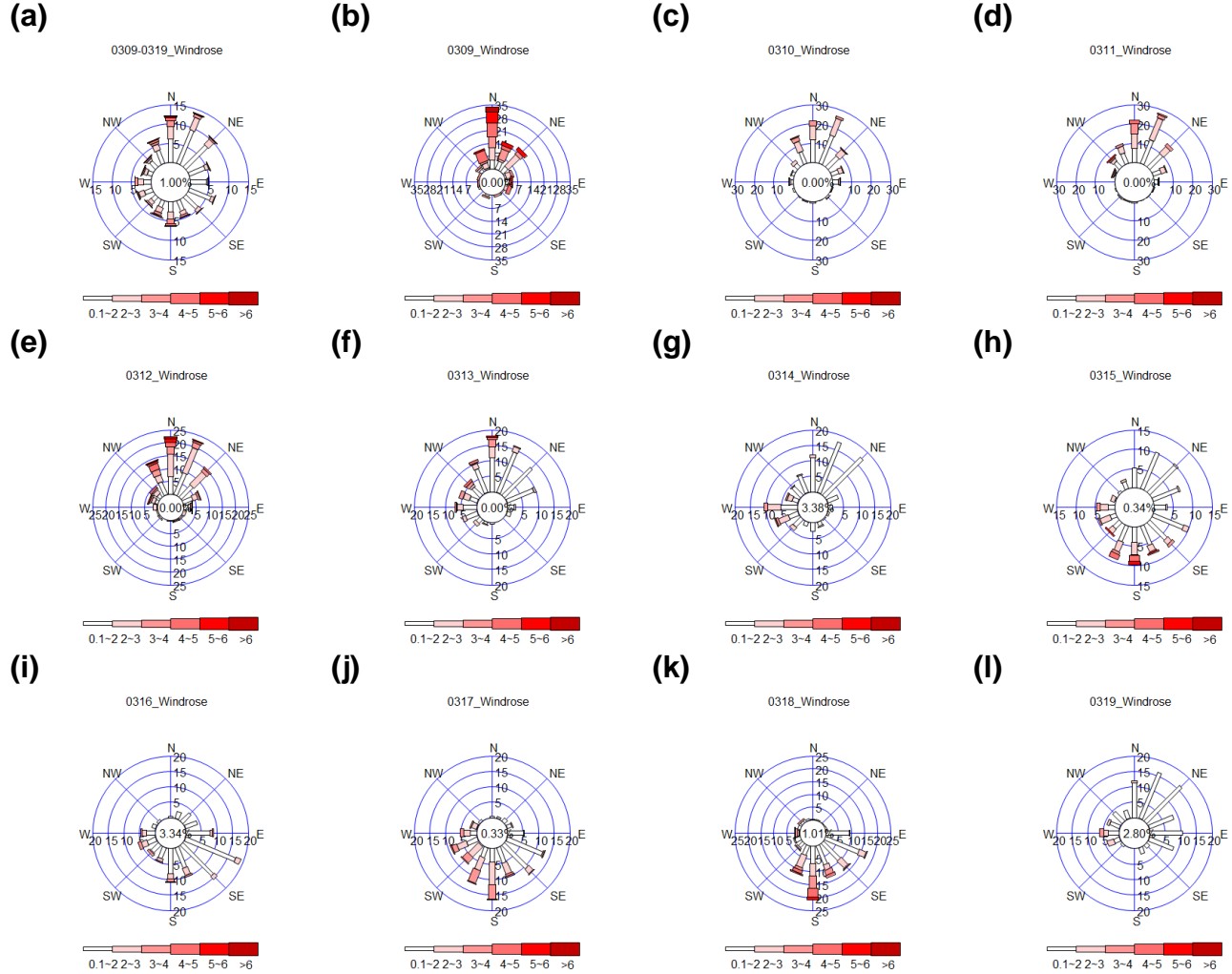

**Figure 3: Windroses for (a) 2015/3/9–3/19, (b to l) each day during the 2015/3/9–3/19 period, respectively; 2015/3/9–3/14 from the NNW–N–NNE direction; 2015/3/15–3/18 from the SSW–S–SSE–SE–ESE direction**

### Acetone

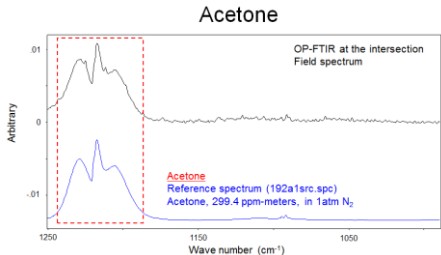

### Ammonia

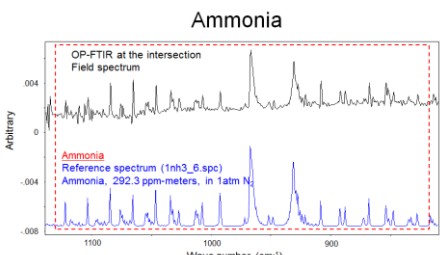

### m-Xylene

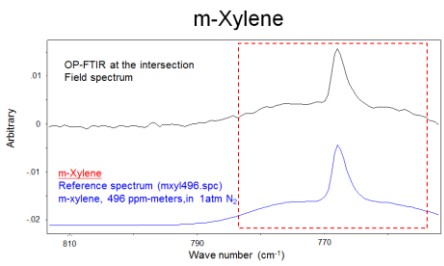

### o-Xylene

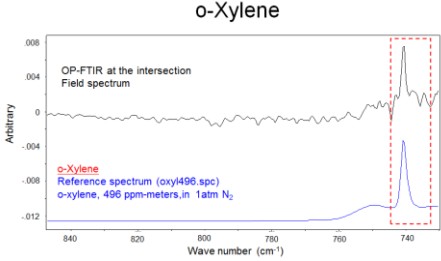

### Toluene

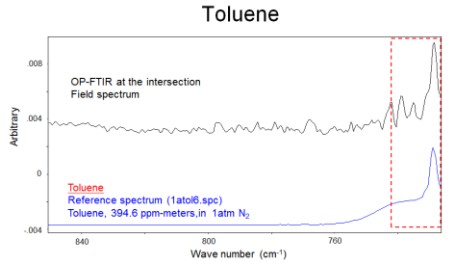

### Ethyl acetate

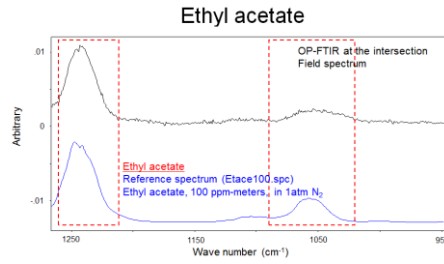

### Gasoline

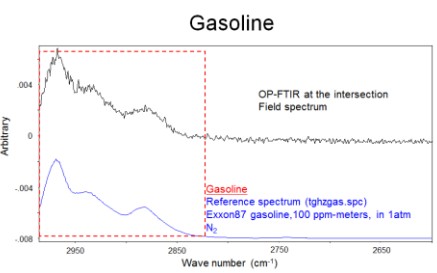

### Nitrogen dioxide

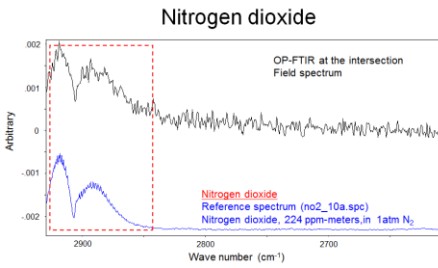

### n-Butyl Acetate

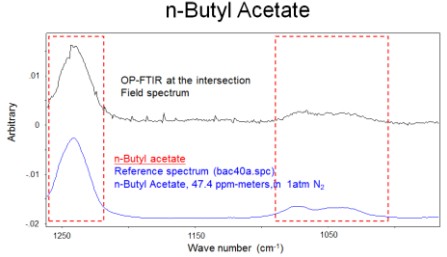

### Propylene Glycol Methyl Ether Acetate

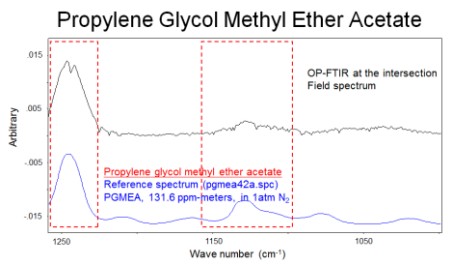

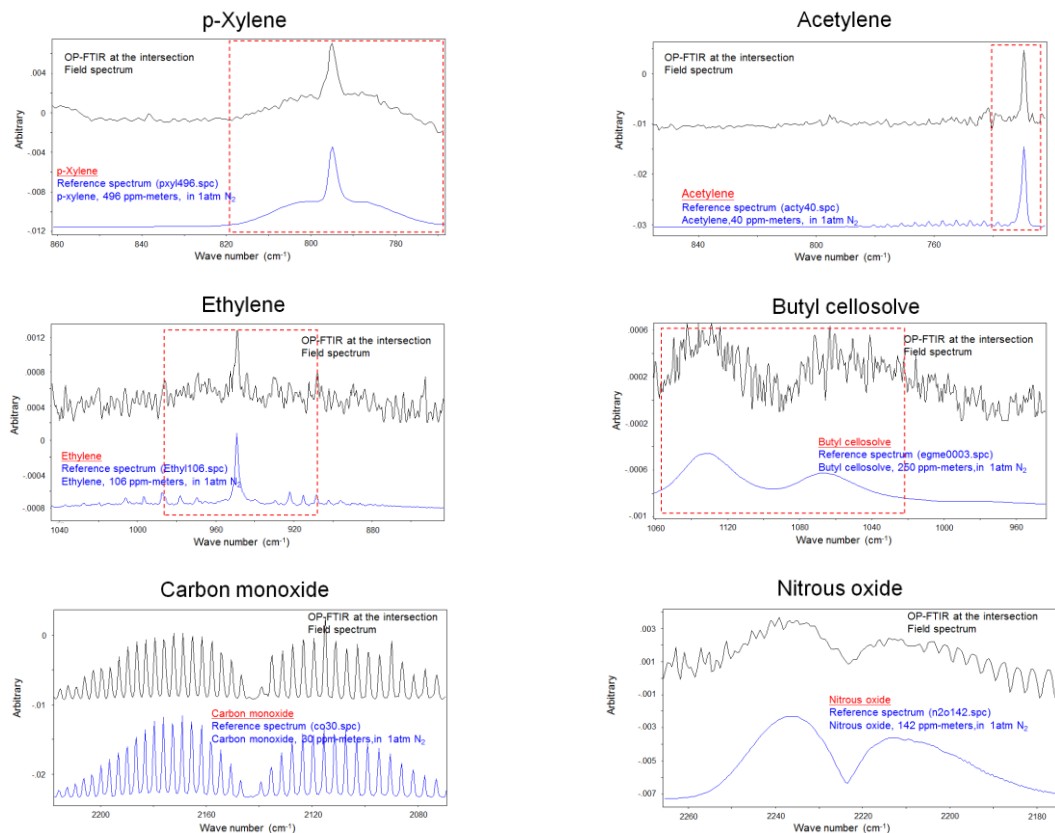

**Figure 4: Comparison between measured spectra (at the receptor site) and reference spectra (from the spectra library)**

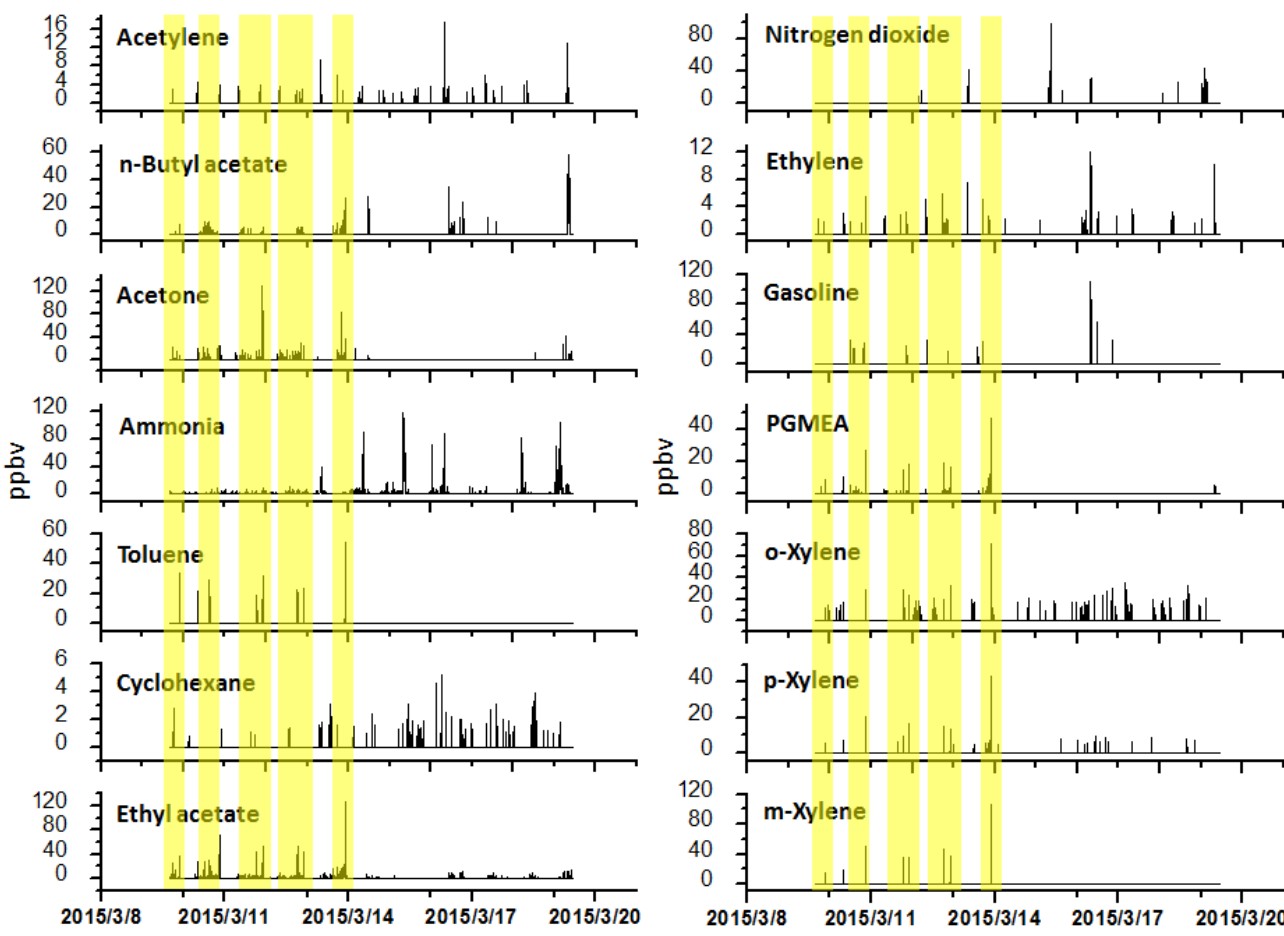

**Figure 5: Time-series pattern of chemical species detected at the receptor site by OP-FTIR; the yellow highlights indicated the periods when the odor was reported**

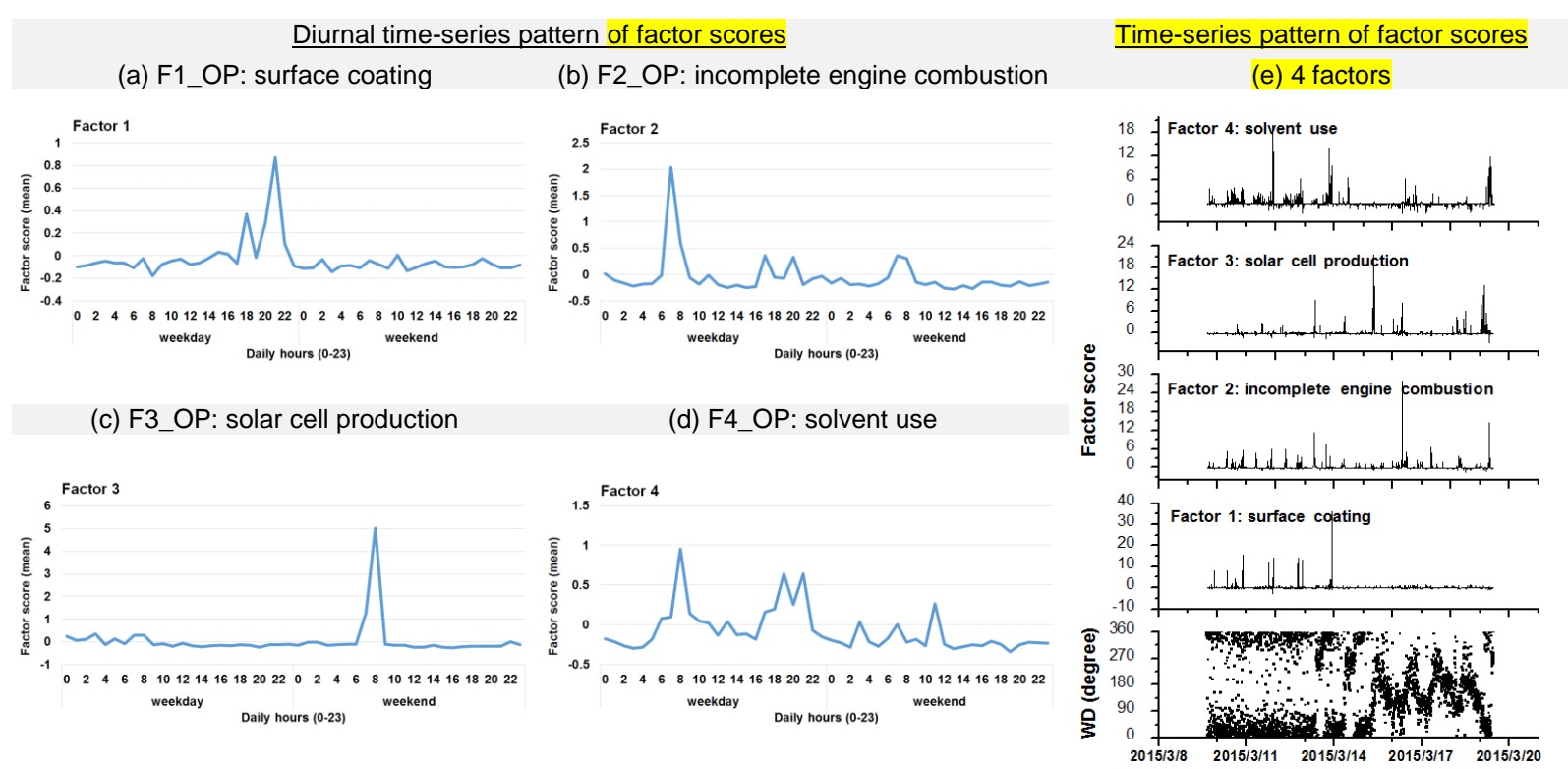

**Figure 6: Diurnal time-series pattern of factor scores for the four factors/sources categorized by OP-FTIR at the receptor path; (a) first group (F1_OP): surface coating; (b) second group (F2_OP): incomplete engine combustion; (c) third group (F3_OP): solar cell production; (d) fourth group (F4_OP): solvent use; (e) time-series pattern of factor scores of the four factors, suggesting that the proportion of the factor scores in negative values were in a relatively small range.**

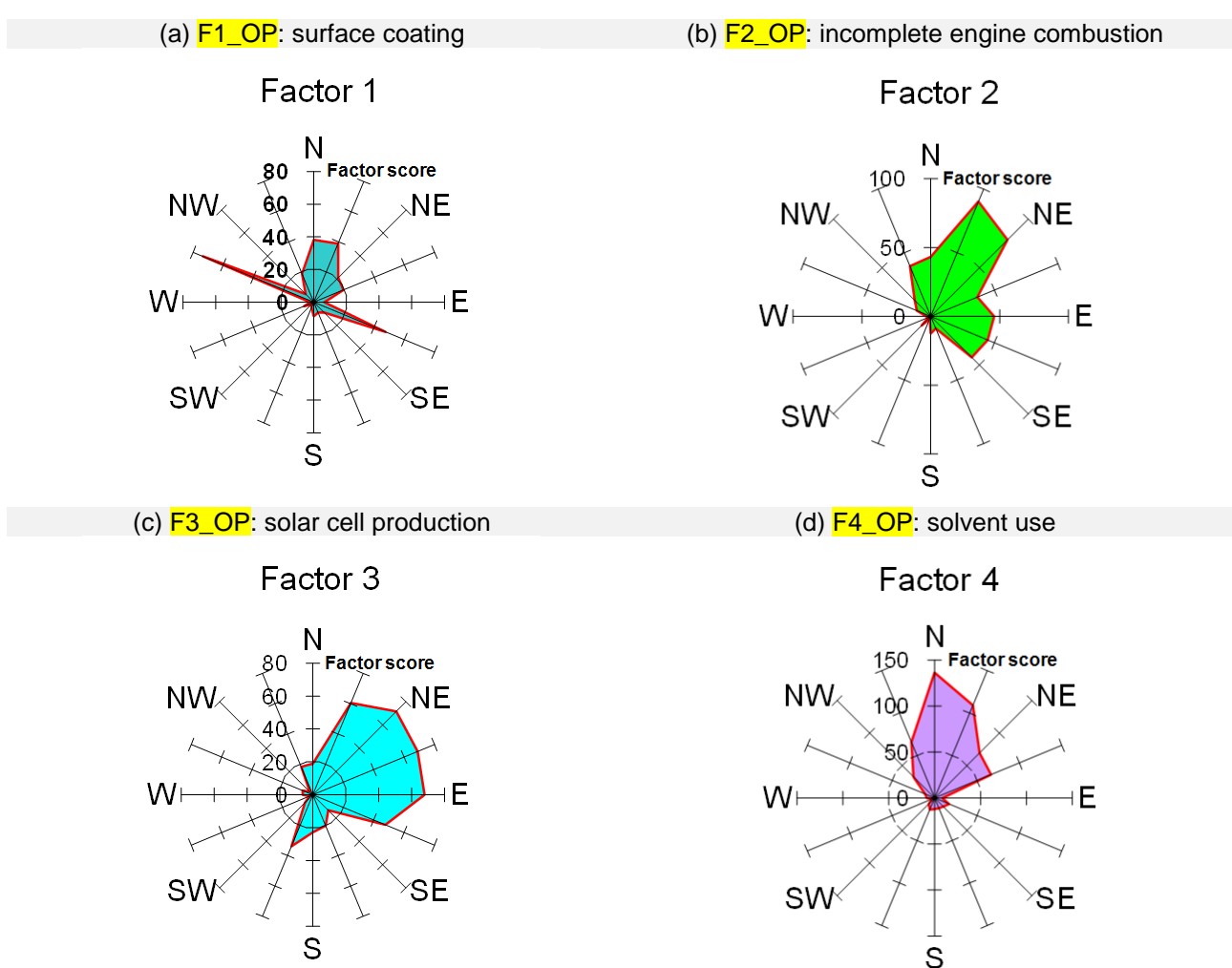

(a) F1_OP: surface coating

(b) F2_OP: incomplete engine combustion

(c) F3_OP: solar cell production

(d) F4_OP: solvent use

**Figure 7: Windrose diagrams of factor scores for the four factors/sources categorized by OP-FTIR at the receptor path; (a) first group (F1_OP): surface coating; (b) second group (F2_OP): incomplete engine combustion; (c) third group (F3_OP): solar cell production; (d) fourth group (F4_OP): solvent use**

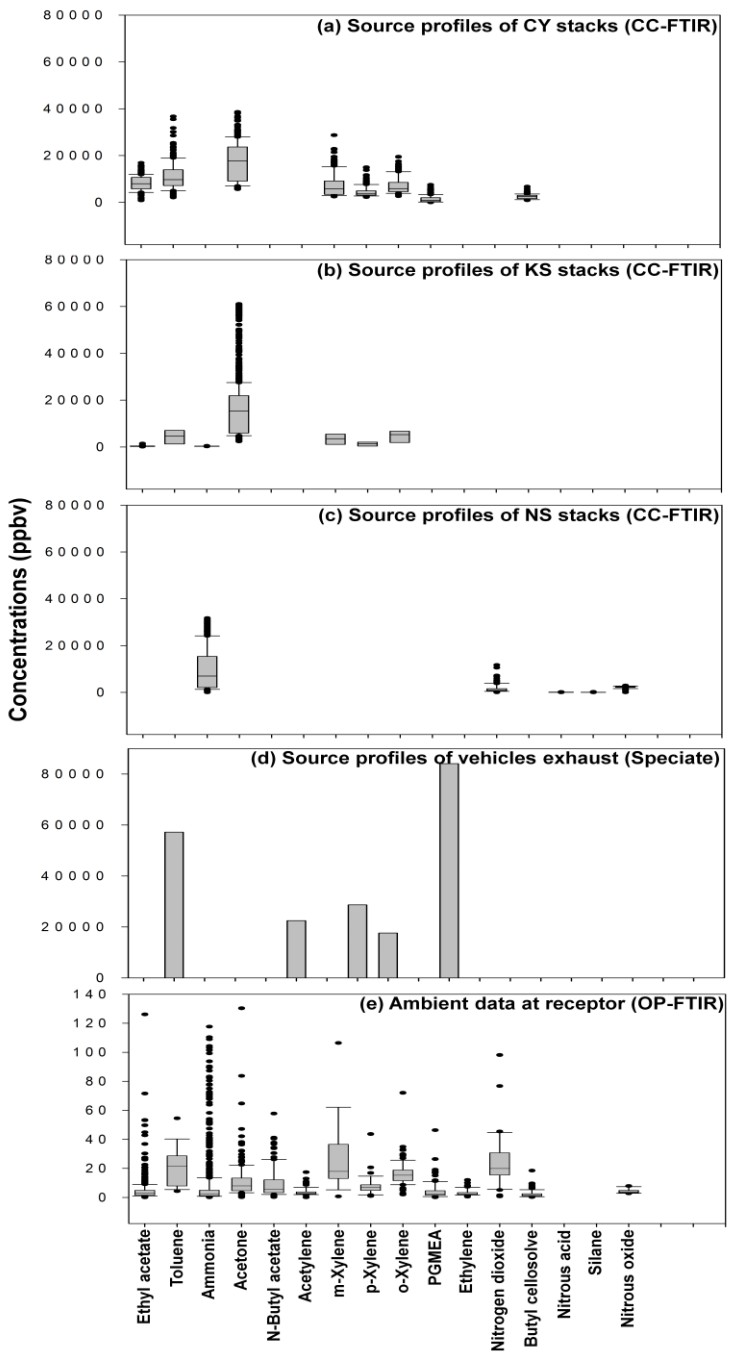

**Figure 8: Panel plots showing relationships between source profiles and ambient data: (a) CY source profile; (b) KS source profile; (c) NS source profile; (d) traffic source profile; (e) ambient data at the receptor**

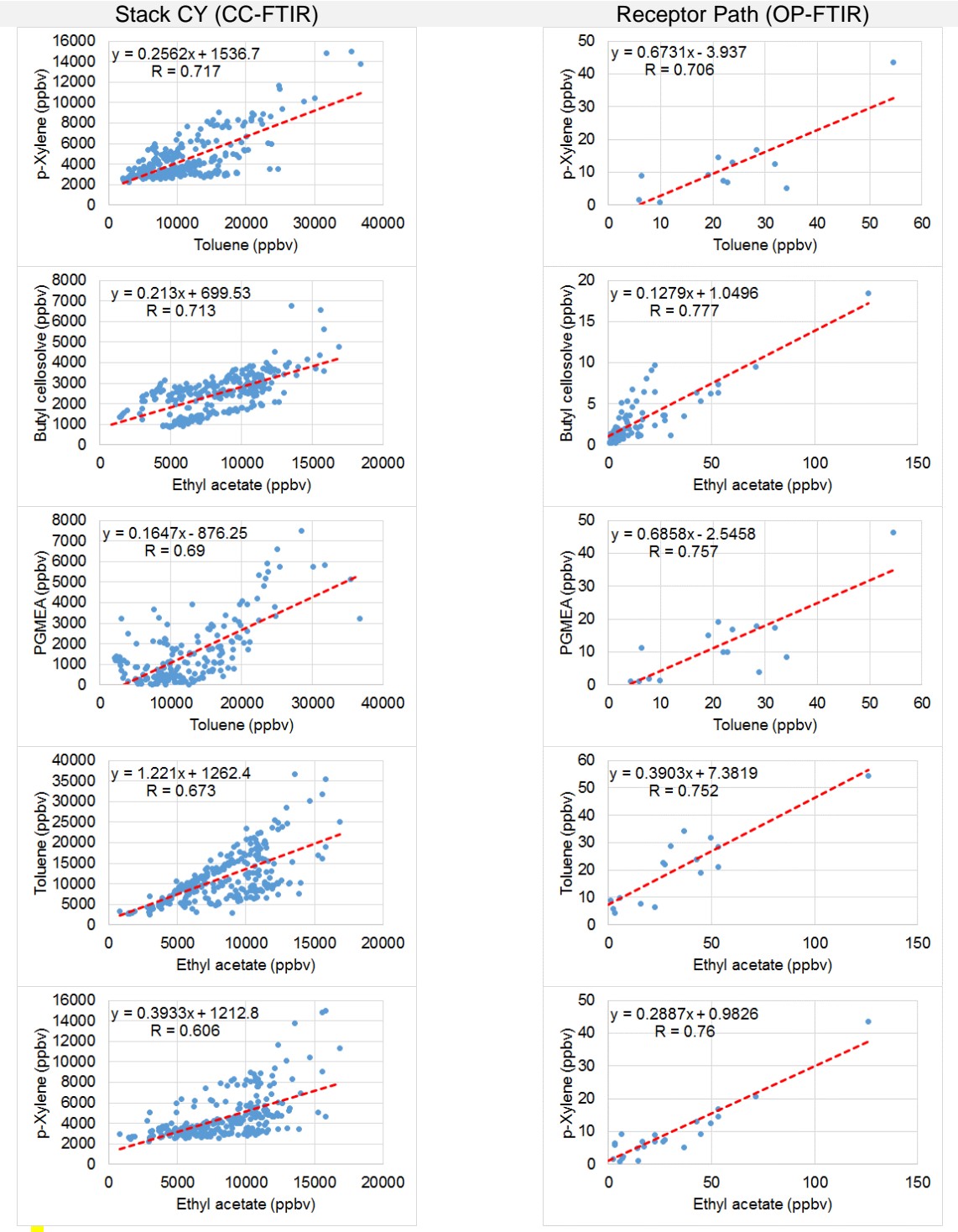

**Figure 9: Scatter plots of concentration variations over time between two detected contaminants from CY stacks (CC-FTIR) and receptor path (OP-FTIR). The correlation coefficients were mostly greater than 0.7.**

## 14. Graphical abstract

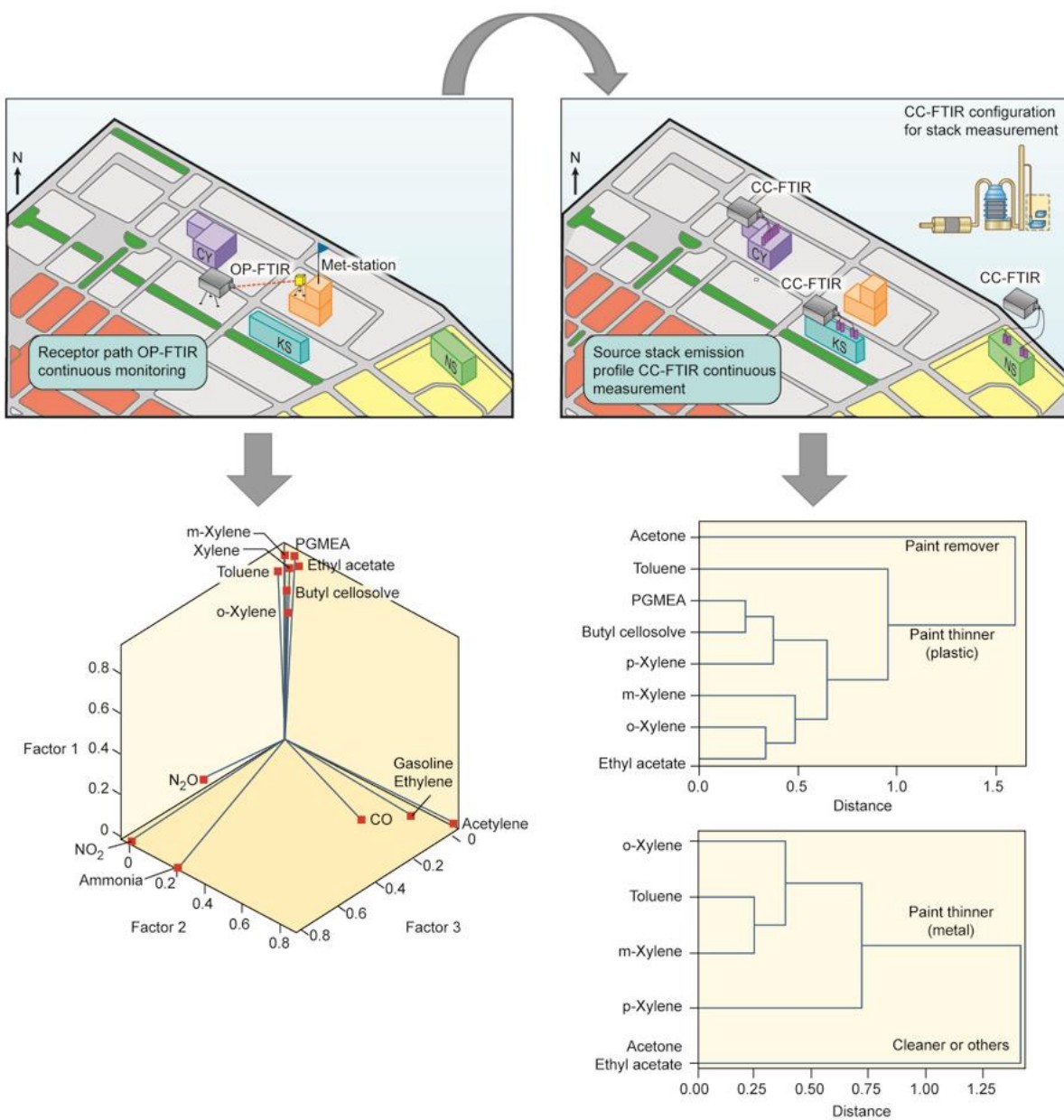

**Figure A1: Graphical abstract illustrating the concept of using a dual-optical sensing system to generate receptor and source continuous monitoring data for performing the qualitative source apportionment of factor and cluster analyses in this study**