# Peer review of "Application of Factor and Cluster Analyses to Determine Source– Receptor Relationships of Industrial Volatile Organic Odor Species in a Dual-Optical Sensing System"

_Atmospheric Measurement Techniques, 2018_

## Referee Comment (RC1) · Anonymous Referee #1 · 15 Apr 2019

The study "Application of Factor and Cluster Analyses to Determine Source– Receptor Relationships of Industrial Volatile Organic Odor Species in a Dual-Optical Sensing System " by Jen-Chih Yang et al. uses FTIR spectroscopic measurements to determine the source of the emission of Volatile Organic Odor Species near an industrial park in southern Taiwan. The measurements are carried out at three sites (CY: a sun glass factory, KS: a light metal casing factory, NS: solar cell manufacturer), where the odor and contamination might be emitted and another where the odor and the related species are detected. Therefore in this study the composition of the stack emissions

from three different industrial complexes are characterized using closed cycle FTIR-spectroscopic measurements and at another time the pollution event with its characteristic odor species using open path FTIR spectroscopy using a retroreflector and 2x143m pathl ength . FTIR spectroscopy allows the determination of the concentration from various gases and is very suitable for this purpose.

Here the technique in both geometries (closed cell measurements and open path) might allow according to the authors the determination of more than 300 pollutant species using the commercial software based on linear least square fitting. At least 17 species are in deed detected and reported in table A1 and figure A3 of this study. The contribution of the different sources to the pollution events are identified by their specific composition, emission profile or here named Factors $F\_i$. Therefore different industrial processes are related with a specific combination of different pollutants. This combination are obtained using the two statistical method: Factor analysis and Cluster analysis.

The technique which allows for the simultaneous measurement of different gases is very suitable for the "Factor analysis" and the determination of different emissions profiles produced by different processes and the Cluster analysis as well the meteorological data confirm these findings.

The approach which allows to identify the different sources and determine sus contributions to the odor quantitatively is interesting, meets the scope of the journal, is in general well written and therefore suitable for publication in AMT after some revision in more detailed described below and after providing some more detailed information

General: The strength of the article is the nice an clear concept using FTIR measurements which allows for the measurement of 17 species simultaneously, measure "emission profiles " of these species at the potential sources using cell measurements and detect these emission profiles at the receptor site using open path measurements and compensate the dilution using an increased path length. The statistical methods

[Figure]

are chosen in a proper way and the results are clear. In addition the study also is able to determine different industrial processes, which are occurring at the three odor producing sites, and the meterological conditions, which confirm the identification of the origin of the odor of different events. The structure of the article might be logical, but the references to the figures in the text is sparse and might be missing sometimes and the use of T1 and TA1 as well as Figure 1 y Figure A1, without adding an Appendix with a text document is confusing and make it a bit difficult to understand the article. Therefore work is required to improve the manuscript and its readability before publication.

There are some questions, which are not easily to answer, but it should be clear for the reader: 1.) Two set up are used but to my understanding it would work also with just one FTIR, as the measurements at the potential sources are realized independently and also the open path measurements at the two sites do not have to be simultaneously. 2.) Factor and cluster analysis are two independent methods: Are they used to confirm each other, compliment some aspects or as "combined" analysis 3.) Which role-plays the meteorology in the analysis? The meteorological data might also be used in the cluster analysis or are only used to confirm the other method in the different cases. Especially here you use the different factors to confirm that the method works proper and you can get the sources from the factors.

0. Abstract: Maybe the author could already state in the abstract the location and time where the measurement where taken. This study developed and alternative approach and an dual. . .. Air contaminants in and nearby a industrial park in southern Taiwan.

line 6 page 2 "Continuous monitoring" alone is a bit misleading, if you talk about both receptor site and source emissions, because it implicitly say "simultaneously", which is not the case. Maybe you can add just the periods where sources and where receptor sites are "measured continuously". Or just add "during different periods"

2. Materials and Methodology:

1.1 Site description and sampling techniques Page 4, line 14-page 5 line 19

I would recommend to separate "Site description" line 16-26 and "sampling techniques" line 27-19

Please could you add a map with all sites and distances.

Measurement method:

The end of the introduction already gives description of the method, for me that is ok, but please add there or here the very important information about the path length in the "Closed Cell" the used pressure in the cell and a estimation of the gas flow through the cell. Did you try to reduce the water vapour to decrease interference with H2O absorption or is it not necessary.

2.2Chemical analysis methods

There is no always a link to the figures and tables, but from the table 1 it can be seen that you found 17 relevant species, this is more interesting than the list of species in the library. I would recommend move the parts of the description of the technique from the Instrument manual to the introduction (e.g. more than 300 species) and the concrete chosen settings and the indeed used species in the method- section. It would be nice to get the complete set of 17 micro windows, where the 17species are retrieved also in a Table as Figure 3A has very small characters at the x axes and shows only 16 species.

How the rolling background is calculated and used should be explained more detailed.

2.3 Qualitative receptor modelling: It is not very clear described. What is the index in X_?, X_1, X_2 in Eq 1-4. Or the first index in the coefficient a11. I would assume the index X_1= X_t1 and describe the time of the observations, as the origin of the Factors F1,... is not stated it seems to be taken from the OP-observations. I would like something like F1_op (open path 1) and maybe an corresponding Factor F1_CY o F1_NS , F1_KS. So it is very clear. And please report the factors at the receptor site

and the source sites in an comparable way. Maybe complete the factors in table 3 as they are in table 2 just by adding 0.0. And please add the Factors from the site NS to table 3.

Maybe you could also report a table with the scalar-products of <F1_CY , F2_OP> for the 4 open path factors Fi_OP with all source factors. Just taking Table 2 and table 3 (after adding the NS factors) would be a 4x(2+2+factors NS) Table/Matrix. It would even be interesting, if the factors of the different sources NS,CY,KS are more or less orthogonal or have a strong overlap.

3.2 Ambient data from receptor path:

Please add a Figure with the time-series which are the basis for the calculation of r_phi and r_pb, OP-FTIR measurements and indicate, when odor was reported. Figure 2:shows no values, when the factor might be negative, please correct it, even if the contribution of the factor is negative it has to be reported. There should be errors in the coefficients , which explain negative values as least in a small range. Caption Figure 2: Time-series pattern -> Diurnal time-series pattern

Missing Figure : Could you add a complete time series of the 4 factors found at the receptor site, which is not a diurnal pattern

4.Conclusions: Is a bit short and very arbitrary and a little redundant.

p.10 l.17: I would replace "dual-optical sensing system" by "FTIR- spectroscopic measurements." And clarify less ambivalent how the meterological data and cluster analysis was used. Maybe something similar as: "This study developed an alternative investigative framework for detecting air pollution sources of odor nuisance by measuring 17 gas species simultaneously using FTIR spectroscopic measurements and factor analyses to identify and characterize emission sources of multiple air contaminants. Meteorological data and Cluster analyses were employed to proof the identification of the major odor emissions" Maybe you could add some numbers how offen the odor

occures which originate from CY,KS,NS and the different processes.

Figures and Table: Please do not use Appendix if you refer to the figure or table in the main text and ensure that all figures and tables are mentioned in the text and keep the order how they are used in the text.

---

## Author Comment (AC1) · 11 Jun 2019

Response to Interactive discussion: 'amt-2018-344', Anonymous Referee #1, 15 April 2019

Q1: General comments: The simultaneous measurement of different gases is very suitable for the "Factor analysis" and the determination of different emissions profiles produced by different processes; the "Cluster analysis" as well as the meteorological data confirm these findings. This approach which allows to identify the different

sources and determine contributions to the odor quantitatively is interesting, meets the scope of the journal, is in general well written and therefore suitable for publication in AMT after some revision in more detailed described below and after providing some more detailed information. The strength of the article is the nice an clear concept using FTIR measurements which allows for the measurement of 16 species simultaneously, measure "emission profiles " of these species at the potential sources using cell measurements and detect these emission profiles at the receptor site using open path measurements and compensate the dilution using an increased path length. The statistical methods are chosen in a proper way and the results are clear. In addition the study also is able to determine different industrial processes, which are occurring at the three odor producing sites, and the meterological conditions, which confirm the identification of the origin of the odor of different events. The structure of the article might be logical, but the references to the figures in the text is sparse and might be missing sometimes and the use of T1 and TA1 as well as Figure 1 y Figure A1, without adding an Appendix with a text document is confusing and make it a bit difficult to understand the article. Therefore work is required to improve the manuscript and its readability before publication. 1.) Two set up are used but to my understanding it would work also with just one FTIR, as the measurements at the potential sources are realized independently and also the open path measurements at the two sites do not have to be simultaneously.

Reply: Thanks for the comment. Yes, the measurement at emission sources would work with only one set of CC-FTIR. Also the OP-FTIR measurement does not have to be operated simultaneously at the two sites as well.

2.) Factor and cluster analysis are two independent methods: Are they used to confirm each other, compliment some aspects or as "combined" analysis

Reply: Thanks for the comment. Factor and cluster analysis are used to confirm each other. The concurrent trends between different species measured by the CC-FTIR can be analyzed using both factor and cluster analysis. In order to gain insight into the

underlying emission source characteristics, odor contaminants with concurrent patterns were grouped together as a factor. Cluster dendrograms provide linkage paths between groups of chemicals to offer more information about the characteristics of different emission sources.

3.) Which role-plays the meteorology in the analysis? The meteorological data might also be used in the cluster analysis or are only used to confirm the other method in the different cases.

Reply: Thanks for the comment. A meteorological station was operated simultaneously with an OP-FTIR system to collect continuous wind data that can enable identification of the incoming direction of different odor contaminants, and provide spatiotemporal measurement of odor pollutants. In other words, the meteorological data was used to confirm the factor analysis in the way that the incoming direction of each factor (representing a group of chemicals) may be different according to the locations of each potential odor sources.

Q2: 0. Abstract : Line 6 page 2 "Continuous monitoring" alone is a bit misleading, if you talk about both receptor site and source emissions, because it implicitly say "simultaneously", which is not the case. Maybe you can add just the periods where sources and where receptor sites are "measured continuously". Or just add "during different periods"

Reply: Thanks for the comment. The phrase "continuous monitoring" in Page2 (Line 6) was rewritten as "Both receptor and source monitoring data were collected to characterize the emission sources of various odorous substances".

Q3: 2. Materials and Methodology (2.1 Site description and sampling techniques): 1.) Page 4, line 14-page 5 line 19. I would recommend to separate "Site description" line 16-26 and "sampling techniques" line 27-19. Please could you add a map with all sites and distances?

Reply: Thanks for the comment. Page4 (Line 16-26) has been separated from Page4 (Line 27) -Page5 (Line 19), and the subtitles were renamed subsequently: 2.1 site description; 2.2 sampling techniques; 2.3 chemical analysis methods; and 2.4 qualitative receptor modeling. A map with all sites and distances was added in Figure 2.

2.) Measurement method: the end of the introduction already gives description of the method, for me that is ok, but please add there or here the very important information about the path length in the "Closed Cell" the used pressure in the cell and an estimation of the gas flow through the cell. Did you try to reduce the water vapour to decrease interference with H2O absorption or is it not necessary.

Reply: Thanks for the comment. The following sentence has been added to Page 5 (Line 14-18) –"A 10-m (path length) gas cell with the inner pressure of 720 mm Hg, and an estimated gas flow rate of 0.37 Liter/Sec. was used for the CC-FTIR multi-reflection gas measurements. The water vapour was mostly removed by using an impinger connected to the inlet of the gas cell to decrease interference with H2O absorption in the FTIR spectra".

Q4: 2. Materials and Methodology - 2.2 Chemical analysis methods: 1.) There is no always a link to the figures and tables, but from the table 1 it can be seen that you found 17 relevant species, this is more interesting than the list of species in the library. I would recommend move the parts of the description of the technique from the Instrument manual to the introduction (e.g. more than 300 species) and the concrete chosen settings and the indeed used species in the method- section. It would be nice to get the complete set of 17 micro windows, where the 17species are retrieved also in a Table as Figure 3A has very small characters at the x axes and shows only 16 species.

Reply: Thanks for the comment. The following sentences – "The IR "fingerprints" of over 300 compounds were established on the basis of information from the US Environmental Protection Agency (USEPA) and the FTIR software developers" has been

moved from Page 5 (Line 26-28) to the introduction section on Page 3 (Line 29-30). The following sentences were added to the method section – "The unique fingerprint characteristics of each chemical compound brought identification of gaseous pollutants possible through comparing the shape, position and relative peak height of each measured spectrum with reference spectra." on Page 5 (Line 32) - Page 6 (Line 2). The sentence "any gaseous compounds absorbed in the IR region (approximately 2.5–25 microns) were potential candidates for monitoring using FTIR technology" originally on Page 5 (Line 25-26) was moved to Page 5 (Line 28-29) for a better explanation of the analytical techniques. All characters in the x and y axes have been enlarged and all 16 species have been included in the Figure 4.

2.) How the rolling background is calculated and used should be explained more detailed.

Reply: Thanks for the comment. The rolling background was collected using the first spectrum as a background to create an absorbance spectrum from the second spectrum, using the second spectrum as a background for the third spectrum and so on. The integral values of concentrations are calculated to obtain time series data for each compound. The advantage of using the rolling background is that it will have the best correction for water vapor, detector and instrument response, and the lowest residual error.

Q5: 2. Materials and Methodology - 2.3 Qualitative receptor modelling: 1.) It is not very clear described. What is the index in $X\_?$, $X\_1$, $X\_2$ in Eq 1-4. Or the first index in the coefficient a11. I would assume the index $X\_1 = X\_t1$ and describe the time of the observations, as the origin of the Factors F1,...is not stated it seems to be taken from the OP-observations. I would like something like F1_op (open path 1) and maybe an corresponding Factor F1_CY o F1_NS , F1_KS. So it is very clear. And please report the factors at the receptor site and the source sites in a comparable way. Maybe complete the factors in table 3 as they are in table 2 just by adding 0.0. And please add the Factors from the site NS to table 3.

[Figure]

Reply: Thanks for the comment. In order to explain the meaning of each index in Eq 1-4 $X_1 = a_{11}f_1 + a_{12}f_2 + \ldots + a_{1m}f_m + e_1$ Eq. (1) $X_2 = a_{21}f_1 + a_{22}f_2 + \ldots + a_{2m}f_m + e_2$ Eq. (2) $X_p = a_{p1}f_1 + a_{p2}f_2 + \ldots + a_{pm}f_m + e_p$ Eq. (3) $X = (X_1,\ldots,X_P)'$, $f = (f_1,\ldots f_m)'$, and $e = (e_1,\ldots e_p)'$ Eq. (4) The following sentences have been added to Page 6 (Line 13-15) – "where $X_i$ = the ith chemical species with mean 0 and unit variance, $i = 1,\ldots,p$; $a_{i1}$ to $a_{im}$ = the factor loadings for the ith chemical species; $f_1$ to $f_m$ = m uncorrelated common factors, each with mean 0 and unit variance; e = the error terms indicating the residual part of $X_i$ that is not in common with the other variables". In Eq 1-4, "$a_{11}$" represents the factor loading of factor 1 ($f_1$) for the first chemical species, "$a_{12}$" represents the factor loading of factor 2 ($f_2$) for the first chemical species; "$a_{21}$ represents the factor loading of factor 1 ($f_1$) for the second chemical species. $X_1$ describes the communality or common variance of the first chemical species; whereas, $X_2$ represents the communality or common variance of the second chemical species. The factor # based on OP-FTIR (e.g. F1_OP) and the factor # based on CC-FTIR with its corresponding source name (e.g. F1_CY, F1_NS, F1_KS) have been added to Table 2, Table 4, Figure 6 and Figure 7; the factors at the receptor site and the source sites are now presented in a comparable way. The factors from the NS stacks has been added to table 4 (originally labeled as table 3) on Page 17 and the following sentences were added to Page10 (Line 12-14) – "The chemicals from the NS stacks were mainly inorganic materials (nitrous oxide, silane, ammonia, nitrous acid, and nitrogen dioxide) that were commonly used in the solar cell production (Table 4e), all of which were not corresponded with the organic odorous solvents identified in the receptor sites".

2.) Maybe you could also report a table with the scalar-products of <F1_CY , F2_OP> for the 4 open path factors Fi_OP with all source factors. Just taking Table 2 and table 3 (after adding the NS factors) would be a 4x (2+2+factors NS) Table/Matrix. It would even be interesting, if the factors of the different sources NS,CY,KS are more or less orthogonal or have a strong overlap.

Reply: Thanks for the comment. A scalar product is a scalar value that is the result of

an operation of two vectors with the same number of components. Given two vectors A and B each with n components, the scalar product is calculated as:

$$A \cdot B = A_1B_1 + ... + A_nB_n \quad .Eq.A1$$

The scalar-products of 4 open path factors (table 2) with all source factors [table 4 (originally labeled table 3)] were calculated using Eq. A1. In order to perform the calculation of scalar –products, the factor loadings (in table 2 & 4) were replaced by the eigenvectors generated by the SAS programs. The outcomes of a 4 x 6 table/matrix was shown in table A below(see also the attached 'Table A' on the last page of this document). It is suggested that none of the scalar-products was orthogonal except for the one calculated from F4_OP $\cdot$ F2_NS, indicating that F4_OP was absolutely irrelevant with F2_NS, meaning that F4_OP –"solvent use for paint remover" was absolutely irrelevant with F2_NS–"HNO3 thermal decomposition". The top 3 scalar-products were calculated from F1_OP $\cdot$ F1_CY, F1_OP $\cdot$ F1_KS, and F4_OP $\cdot$ F2_KS with scalar-product values of 0.963, 0.727 and 0.570, respectively; indicating that the angles between these three pairs of vectors were far less than 90 degrees, meaning that the relationships between these three pairs of factors were strongest following the order of F1_OP $\cdot$ F1_CY > F1_OP $\cdot$ F1_KS > F4_OP $\cdot$ F2_KS in comparison with the rest of the pairs in this 4 x 6 table/matrix. Therefore, it would suggest that F1_OP–"paint thinner for surface coating" was highly related to F1_CY–"plastic paint thinner"; whereas F1_OP–"paint thinner for surface coating" was closely related to F1_KS–"metal paint thinner"; F4_OP –"solvent use for paint remover" was also related to F2_KS–"cleaner or others". The results of the scalar-products demonstrated that the factors at both the receptor site and the source sites were intercomparable, which were consistent with the findings in the previous sections of this manuscript.

Table A: The scalar-products of 4 open path factors with all source factors (a 4 x 6 matrix) F1_OP $\cdot$ FI_CY F1_OP $\cdot$ F2_CY F1_OP $\cdot$ FI_KS F1_OP $\cdot$ F2_KS F1_OP $\cdot$ FI_NS F1_OP $\cdot$ F2_NS 0.963 -0.130 0.727 0.303 0.003 -0.002 F2_OP $\cdot$ FI_CY F2_OP $\cdot$ F2_CY F2_OP $\cdot$ FI_KS F2_OP $\cdot$ F2_KS F2_OP $\cdot$ FI_NS

F2_OP Âů F2_NS -0.072 0.001 -0.076 0.005 -0.179 0.074 F3_OP Âů Fl_CY F3_OP Âů F2_CY F3_OP Âů Fl_KS F3_OP Âů F2_KS F3_OP Âů Fl_NS F3_OP Âů F2_NS 0.026 -0.026 0.040 -0.044 -0.309 0.344 F4_OP Âů Fl_CY F4_OP Âů F2_CY F4_OP Âů Fl_KS F4_OP Âů F2_KS F4_OP Âů Fl_NS F4_OP Âů F2_NS -0.092 0.349 -0.310 0.570 0.029 0.000 Note: (1) If A and B are orthogonal (at 90 degrees to each other), the result of the scalar product will be zero; (2) If the angle between A and B are less than 90 degrees, the scalar product will be positive (greater than zero); (3) If the angle between A and B are greater than 90 degrees, the scalar product will be negative (less than zero)

Q6: 3. Results and Discussion - 3.2 Ambient data from receptor path: 1.) Please add a Figure with the time-series which are the basis for the calculation of r_phi and r_pb, OP-FTIR measurements and indicate, when odor was reported. Figure 2: shows no values, when the factor might be negative, please correct it, even if the contribution of the factor is negative it has to be reported. There should be errors in the coefficients, which explain negative values as least in a small range. Caption Figure 2: Time-series pattern -> Diurnal time-series pattern. Missing Figure: Could you add a complete time series of the 4 factors found at the receptor site, which is not a diurnal pattern.

Reply: Thanks for the comment. The time-series pattern of chemical species (used as the basis for the calculation of r_phi and r_pb) detected at the receptor site by the OP-FTIR has been added in Figure 5; the yellow highlights in the figure indicated the periods when odor was reported. The following sentences were added to Page 8 (Line 7-9) – "A complete time series pattern of chemical species found at the receptor site that were used as the basis for the calculation of r_phi and r_pb was shown in Fig. 2, in which the periods when odor was reported were highlighted". The negative values in Figure 6a to 6d (originally labeled as Figure 2a to 2d) have been added to the diurnal time-series trends. Caption Figure 6 (originally labeled as Figure 2): "Time-series pattern" has been revised as "Diurnal time-series pattern". A complete time series pattern (not diurnal pattern) of the four factors found at the receptor site has

been added in Figure 6e, which suggested that the proportion of the factor scores in negative values were in a relatively small range.

Q7: 4. Conclusions: Is a bit short and very arbitrary and a little redundant: 1.) p.10 line.17: I would replace "dual-optical sensing system" by "FTIR- spectroscopic measurements." And clarify less ambivalent how the meterological data and cluster analysis was used. Maybe something similar as: "This study developed an alternative investigative framework for detecting air pollution sources of odor nuisance by measuring 17 gas species simultaneously using FTIR spectroscopic measurements and factor analyses to identify and characterize emission sources of multiple air contaminants. Meteorological data and Cluster analyses were employed to proof the identification of the major odor emissions" Maybe you could add some numbers how often the odor occurs which originate from CY,KS,NS and the different processes.

Reply: Thanks for the comment. "Dual-optical sensing system" has been replaced by "FTIR- spectroscopic measurements" in the conclusion section. The role of meterological data and cluster analysis on Page10 (Line 17-19) were clarified by using the sentences suggested by the referee 1. The overall content of the conclusion has been rewritten as follows: "This study developed an alternative investigative framework for detecting air pollution sources of odor nuisance by measuring 16 gas species simultaneously using FTIR spectroscopic measurements and factor analyses to identify and characterize emission sources of multiple air contaminants. Meteorological data and cluster analysis were employed to proof the identification of the major odor emissions. Different industrial processes were related to a specific combination of different pollutants, and this combination was obtained using the two statistical methods of factor analysis and cluster analyses. Factor and cluster analyses were employed to improve the quality and completeness of the source profiles. A field study used FTIR spectroscopic measurements to determine the source of the emission of volatile organic odor species near an industrial park in southern Taiwan demonstrated the feasibility of this proposed method. The major odor emission source was identified through qualitative source apportionment of factor and cluster analyses. With enhanced efficiency in odor investigation methodology, future emission reduction plans can be developed and overall air quality can be improved".

Q8: Figures and Table: 1.) Please do not use Appendix if you refer to the figure or table in the main text and ensure that all figures and tables are mentioned in the text and keep the order how they are used in the text.

Reply: Thanks for the comment. The "10. Appendix" section has been removed from the manuscript. All tables and figures originally in "10. Appendix section" were rearranged to either "8. Table" or "9. Figure" sections, in which they were reordered according to how they are used in the text to ensure that all figures and tables are mentioned in the text.

Please also note the supplement to this comment:
https://www.atmos-meas-tech-discuss.net/amt-2018-344/amt-2018-344-AC1-supplement.pdf

Table A: The scalar-products of 4 open path factors with all source factors (a 4 x 6 matrix)

| F1_OP · FI_CY | F1_OP · F2_CY | F1_OP · FI_KS | F1_OP · F2_KS | F1_OP · FI_NS | F1_OP · F2_NS |
|---|---|---|---|---|---|
| **0.963** | -0.130 | **0.727** | **0.303** | 0.003 | -0.002 |
| F2_OP · FI_CY | F2_OP · F2_CY | F2_OP · FI_KS | F2_OP · F2_KS | F2_OP · FI_NS | F2_OP · F2_NS |
| -0.072 | 0.001 | -0.076 | 0.005 | -0.179 | 0.074 |
| F3_OP · FI_CY | F3_OP · F2_CY | F3_OP · FI_KS | F3_OP · F2_KS | F3_OP · FI_NS | F3_OP · F2_NS |
| 0.026 | -0.026 | 0.040 | -0.044 | **-0.309** | **0.344** |
| F4_OP · FI_CY | F4_OP · F2_CY | F4_OP · FI_KS | F4_OP · F2_KS | F4_OP · FI_NS | F4_OP · F2_NS |
| -0.092 | **0.349** | **-0.310** | **0.570** | 0.029 | 0.000 |

Note: (1) If A and B are underline{orthogonal} (at 90 degrees to each other), the result of the scalar product will be underline{zero}; (2) If the angle between A and B are underline{less than 90 degrees}, the scalar product will be underline{positive} (greater than zero); (3) If the angle between A and B are underline{greater than 90 degrees}, the scalar product will be underline{negative} (less than zero)

**Fig. 1.**

**Supplement:**

[revised manuscript text omitted]
|---|---|---|---|---|---|---|---|---|---|---|
| Acetone | 182 | 130.3 | 11.61±1.02 | 118.8 | 37 | 1.10 | 0.672 | 0.432 | <0.001*** | <0.001*** |
| Ethyl Acetate | 519 | 126.1 | 4.95±0.38 | 176.5 | 170 | 0.23 | 0.551 | 0.450 | <0.001*** | <0.001*** |
| Ammonia | 982 | 117.6 | 7.07±0.50 | 223.0 | 45 | 0.26 | 0.286 | -0.129 | <0.001*** | 0.046* |
| Gasoline (mixture of C5+ and BTEX) | 22 | 110.2 | 33.21±5.00 | 70.6 | 25 | 0.77 | 0.155 | 0.016 | 0.017* | 0.810 |
| m-Xylene | 17 | 106.4 | 27.96±6.05 | 89.2 | 81 | 1.29 | 0.328 | 0.272 | <0.001*** | <0.001*** |
| Nitrogen Dioxide | 35 | 98.1 | 25.13±3.28 | 77.1 | 58 | 3.08 | -0.161 | -0.107 | 0.013* | 0.099 |
| o-Xylene | 96 | 72.0 | 16.26±0.89 | 53.8 | 180 | 1.12 | -0.001 | 0.027 | 0.987 | 0.680 |
| n-Butyl Acetate | 103 | 57.7 | 9.92±1.05 | 107.2 | 6.3 | 0.28 | 0.345 | 0.020 | <0.001*** | 0.758 |
| Toluene | 16 | 54.5 | 20.57±3.36 | 65.3 | 21 | 1.12 | 0.328 | 0.302 | <0.001*** | <0.001*** |
| Propylene Glycol Methyl Ether Acetate (PGMEA) | 93 | 46.3 | 4.62±0.67 | 139.8 | 25 | 0.20 | 0.569 | 0.312 | <0.001*** | <0.001*** |
| p-Xylene | 39 | 43.6 | 8.15±1.13 | 87.0 | 120 | 0.72 | 0.168 | 0.215 | 0.010* | 0.001** |
| Acetylene | 65 | 17.4 | 3.80±0.35 | 75.1 | 226000 | 0.22 | 0.016 | -0.017 | 0.803 | 0.791 |
| Ethylene | 71 | 12.0 | 3.17±0.25 | 67.0 | 17000 | 0.31 | 0.166 | 0.063 | 0.010* | 0.331 |
| Butyl cellosolve | 137 | 18.5 | 2.30±0.20 | 104.2 | 100 | 0.18 | 0.499 | 0.304 | <0.001*** | <0.001*** |
| Carbon monoxide (above background) | 2779 | | | | Detected | | | | | |
| Nitrous oxide (above background) | 15 | | | | Detected | | | | | |

[revised manuscript text omitted]

**10. Graphical abstract**

[Figure]

**Figure A1: Graphical abstract illustrating the concept of using a dual-optical sensing system to generate receptor and source continuous monitoring data for performing the qualitative source apportionment of factor and cluster analyses in this study**

---

## Referee Comment (RC2) · Anonymous Referee #2 · 9 Jul 2019

Application of Factor and Cluster Analyses to Determine Source–Receptor Relationships of Industrial Volatile Organic Odor Species in a Dual-Optical Sensing System

Jen-Chih Yang, Pao-Erh Chang, Chi-Chang Ho, Chang-Fu Wu

Summary

The authors present the results of a field study carried out over 10 days in March, 2015, whose goals were to identify and characterize odours reported by commuters at an intersection in an industrial park in southern Taiwan. Most likely candidates were

assumed to be industrial VOC emissions from a nearby sunglass factory ("CY") and a metal casings factory ("KS"), which emit organic solvents used in plastic and metal surface coating. Additionally, a nearby solar cell factory ("NS") is a source of inorganic materials used in high-temperature glass sintering.

The experimental approach involved measuring the stack emissions directly with a multi-path closed cell ("CC") connected to an FTIR spectrometer (effectively a point measurement), as well as indirectly at the intersection with an open-path FTIR system (effectively a path-average measurement over 143 m, one way between FTIR spectrometer and retroreflector array). FTIR measurements at all sites were made at 1 cm-1 resolution, averaging 64 IR scans over 5 minutes, and yielding 2,911 consecutive spectra in the case of the open-path system running over 10 days. It is not clear whether the closed cell measurements at the three factories were done at the same time or separately, but there is 1 day of measurements from CY, 10 days from KS and 4 days from NS. Wind speed and direction measurements were also made near the open-path measurement.

Spectral data analysis to derive VOC species concentrations used multicomponent classical least squares with "rolling backgrounds" in the open-path spectra (subject to changing meteorological conditions) and a "fixed reference method" in the closed-cell spectra (which are sampling stack emissions directly).

Subsequent data analysis consisted of performing a factor analysis involving 4 factors (surface coating, incomplete car engine combustion, solar cell production, solvent use) based on the chemical species observed by open-path FTIR at the receptor site ($\sim$16, but there are inconsistencies in the text about which species are being measured and analyzed). The 4 factors (eigenvalues >1) were used in factor loading calculations; chemical species with loadings > 0.4 were considered as influential variables. Factor scores were presented as a function time, separating the one weekend from the remaining weekdays. Additionally, factor scores were presented as windroses.

Species detected at the stacks were compared to ambient open-path data, as well as a database of vehicle exhaust emissions. Finally factor and cluster analysis was performed on measurements inside the stacks, yielding 2 factors with eigenvalues > 1. One of the factors was identical between stack "CY" and the ambient OP-FTIR data Factor 1, leading the authors to conclude that this was the major odour source.

Major Comments

The study is well motivated, the experimental setup and data set is valuable, and the data is presented relatively clearly (e.g., dominant factors in bold/red in tables), however, the paper cannot be published without addressing the comments below.

First, the paper suffers from a number of distracting inconsistencies between what is described in the text and what is found in the tables and figures. For example, chemical species are referred to in the text that are not in in the figures or tables (e.g., cyclohexane, methanol, sulfur hexafluoride, isopropanol, dichloromethane) and the number of species is given as 17 when it is actually 16. Another example involves the discussion of wind directions in Figure 3 regarding OP-FTIR factors, where the named winds do not correspond to the windroses. Another example is a text reference to between 24 and 72 hours of measurements at the stacks but then it states that 2907 spectra were obtained from the KS stack at 5-minute intervals, i.e., a full 10 days of observations.

Second, the study claims to develop an "alternative approach" and to demonstrate the feasibility of an "alternative investigative framework", however, a clear description of the details of the methodology, with references to the literature, is insufficient for an un-initiated reader to make use of this "alternative approach". This applies to 1) the spectral analysis, 2) the factor and cluster analysis of the resultant chemical species concentrations, and 3) the grouping of the data as a function of time and wind direction. The paper is quite short so an expansion is warranted. Figure and table captions are quite sparse, and tables, in particular make improper use of footnote numbering to give information without the number applying to anything in particular in the table (see Table

1 and 2).

Third, there are a number of issues with the spectral data analysis. The detection limits in Table 1 are mostly sub-ppb, which seems high for the OP-FTIR technique in general (see, e.g., Jarvis, 2003, "Open Path Spectrophotometry" in the Instrument Engineer's Handbook), more so given the relatively low number of co-added spectra in a 5 minute period, and the relatively high absolute humidity levels (not given in the paper but inferred). The description of the detection limit calculations in the Table 1 footnote is not clear. Moreover, it is not clear why CO and N2O cannot be retrieved from these FTIR spectra. The issue of "rolling backgrounds" is mentioned, but CO and N2O are far from regions contaminated by water and have been retrieved routinely by others using OP-FTIR, though not with CLS (e.g., Paton-Walsh et al., 2014; Akagi et al., You et al., 2017). Smith et al (2011, AMT) have shown that classical least squares analysis yields inaccurate results at high concentrations (c.f. high concentrations in Table A1 in this study). The nitrous oxide shown in this study in Figure 4, panel (e), has a concentration below 20 ppb (7.9 in Table A1), which is impossible given the ambient concentration of ∼330 ppb. Is the accuracy of other retrieved gases similarly affected?

Fourth, the authors concede that the first factor might not be limited to one source (P8L18), especially when confronted with Figure 3a, which shows high factor scores from NNE, ESE, and also WNW (last direction not discussed though highest factor score from here). Why the sharp features at WNW and ESE as compared to the more spread out feature at NNW,N, NNE, NE? Also, on P8L30 it is stated that six species coexist at both the CY and KS stacks, but only acetone is truly common, the other 5 are emitted ∼1000x more at CY – can that be used to separate them?

Fifth, the authors name CY as the source of the odours based on an identical Factor 1 composition as compared to OP-FTIR. Does this check out with a windrose plot of butyl cellosolve and PGMEA, which are unique to CY based on Table A1? For that matter, can it be verified that when winds blow from stack NS (nearly from the East, not NE as given on P7L34 and P8L1) cyclohexane, acetylene and ethylene increase?

There should also be toluene and xylene. I can understand why no correlation plots (like Figure 5) are shown for KS given that the correlation is discussed as below 0.1, but why is NS not discussed in this way at all?

Sixth, why is Factor 2 (incomplete engine combustion) clustered to the NW and SE (Figure 3b, not discussed), whereas the source directions are indicated as "all directions" in Table 2? It also appears like there are roads all around the open-path in Figure 1. Are some roads more major than others? Seventh, the time-series of factor scores do not correspond to factory working hours in an obvious way. Do the plants run 24 hours a day? Does the solar cell plant run on weekends? Why is there a traffic peak from 6-8 on the weekend? Why does the traffic factor look similar to the solvent factor?

Seventh, the time-series of factor scores do not correspond to factory working hours in an obvious way. Do the plants run 24 hours a day? Does the solar cell plant run on weekends? Why is there a traffic peak from 6-8 on the weekend? Why does the traffic factor look similar to the solvent factor?

---

## Author Comment (AC2) · 5 Aug 2019

Response to Interactive discussion: 'amt-2018-344', Anonymous Referee #2, 09 July 2019. Thank you for allowing us to revise our paper. Attached are our detailed responses to the comments of referee 2. We have attached our replies as Supplement.

Please also note the supplement to this comment:
https://www.atmos-meas-tech-discuss.net/amt-2018-344/amt-2018-344-AC2-supplement.pdf

---

## Author Comment (AC3) · 5 Aug 2019

Response to Interactive discussion: 'amt-2018-344', Anonymous Referee #1, 15 April (second version). Attached are our second version responses to the comments of referee 1. We have attached our second version replies as Supplement.

Please also note the supplement to this comment: https://www.atmos-meas-tech-discuss.net/amt-2018-344/amt-2018-344-AC3-supplement.pdf

---

## Author Response (AR2)

**Title: Application of Factor and Cluster Analyses to Determine Source–Receptor Relationships of Industrial Volatile Organic Odor Species in a Dual-Optical Sensing System**

**Response to Reviewers' Comments**

This document contains a point by point response to referee #1 (from page 1-7) and referee #2 (from page 8-20) of the paper by Jen-Chih Yang et al (amt-2018-344, 2019).

Our responses to the reviewer's comments are in the following format:

- Comments from referees: are shown as **bold** characters.
- Author's response: comments are answered beginning with "Reply".
- Author's changes in the manuscript: If a change was made to the manuscript in response to a comment, the location of that change is highlighted showing the modification in the revised manuscript.

**REFEREE #1:**

**Q1: General comments:**

**The simultaneous measurement of different gases is very suitable for the "Factor analysis" and the determination of different emissions profiles produced by different processes; the "Cluster analysis" as well as the meteorological data confirm these findings. This approach which allows to identify the different sources and determine contributions to the odor quantitatively is interesting, meets the scope of the journal, is in general well written and therefore suitable for publication in AMT after some revision in more detailed described below and after providing some more detailed information.**

**The strength of the article is the nice an clear concept using FTIR measurements which allows for the measurement of 16 species simultaneously, measure "emission profiles" of these species at the potential sources using cell measurements and detect these emission profiles at the receptor site using open path measurements and compensate the dilution using an increased path length. The statistical methods are chosen in a proper way and the results are clear. In addition the study also is able to determine different industrial processes, which are occurring at the three odor producing sites, and the meterological conditions, which confirm the identification of the origin of the odor of different events. The structure of the article might be logical, but the references to the figures in the text is sparse and might be missing sometimes and the use of T1 and TA1 as well as Figure 1 y Figure A1, without adding an Appendix with a text document is confusing and make it a bit difficult to understand the article. Therefore work is required to improve the manuscript and its readability before publication.**

**1.)** **Two set up are used but to my understanding it would work also with just one FTIR, as the measurements at the potential sources are realized independently and also the open path measurements at the two sites do not have to be simultaneously.**

Reply: Thanks for the comment. Yes, the measurement at emission sources would work with only one set of CC-FTIR. Also, the OP-FTIR measurement does not have to be operated simultaneously at the two sites as well.

**2.)** **Factor and cluster analysis are two independent methods: Are they used to confirm each other, compliment some aspects or as "combined" analysis**

Reply: Thanks for the comment. Factor and cluster analysis are used to confirm each other. The concurrent trends between different species measured by the CC-FTIR can be analyzed using both factor and cluster analysis. To gain insight into the underlying emission source characteristics, odor contaminants with concurrent patterns were grouped together as a factor. Cluster dendrograms provide linkage paths between groups of chemicals to offer more information about the characteristics of different emission sources.

**3.)** **Which role-plays the meteorology in the analysis? The meteorological data might also be used in the cluster analysis or are only used to confirm the other method in the different cases.**

Reply: Thanks for the comment. A meteorological station was operated simultaneously with an OP-FTIR system to collect continuous wind data that can enable identification of the incoming direction of different odor contaminants, and provide a spatiotemporal measurement of odor pollutants. In other words, the meteorological data was used to confirm the factor analysis in the way that the incoming direction of each factor (representing a group of chemicals) may be different according to the locations of each potential odor sources.

*Q2: 0. Abstract :*
**Line 6 page 2 "Continuous monitoring" alone is a bit misleading, if you talk about both receptor site and source emissions, because it implicitly say "simultaneously", which is not the case. Maybe you can add just the periods where sources and where receptor sites are "measured continuously". Or just add "during different periods"**

Reply: Thanks for the comment. The phrase "continuous monitoring" in **P2L6-L7** was rewritten as "*Both receptor and source monitoring data were collected to characterize the emission sources of various odorous substances*".

*Q3: 2. Materials and Methodology (2.1 Site description and sampling techniques):*
**1.) Page 4, line 14-page 5 line 19. I would recommend to separate "Site description" line 16-26 and "sampling techniques" line 27-19. Please could you add a map with all sites and distances?**

Reply: Thanks for the comment. **P4L16-26** has been separated from **P4L27 -P5L19**, and the subtitles were renamed subsequently: 2.1 site description; 2.2 sampling techniques; 2.3 chemical analysis methods; and 2.4 qualitative receptor modeling. A map with all sites and distances was added in Figure 2.

**2.) Measurement method: the end of the introduction already gives description of the method, for me that is ok, but please add there or here the very important information about the path length in the "Closed Cell" the used pressure in the cell and an estimation of the gas flow through the cell. Did you try to reduce the water vapour to decrease interference with $H_2O$ absorption or is it not necessary.**

Reply: Thanks for the comment. The following sentence has been added to **P5L14-17** – *"A 10-m (path length) gas cell with the inner pressure of 720 mm Hg, and an estimated gas flow rate of 0.37 Liter/Sec. was used for the CC-FTIR multi-reflection gas measurements. The water vapour was mostly removed by using an impinger connected to the inlet of the gas cell to decrease interference with $H_2O$ absorption in the FTIR spectra"*.

*Q4: 2. Materials and Methodology - 2.2 Chemical analysis methods:*
**1.) There is no always a link to the figures and tables, but from the table 1 it can be seen that you found 17 relevant species, this is more interesting than the list of species in the library. I would recommend move the parts of the description of the technique from the Instrument manual to the introduction (e.g. more than 300 species) and the concrete chosen settings and the indeed used species in the method- section. It would be nice to get the complete set of 17 micro windows, where the 17species are retrieved also in a Table as Figure 3A has very small characters at the x axes and shows only 16 species.**

Reply: Thanks for the comment. The following sentences -- *"The IR "fingerprints" of over 300 compounds were established on the basis of information from the US Environmental Protection Agency (USEPA) and the FTIR software developers"* have been moved to the introduction section on **P3L29-30**. The following sentences were added to the method section -- *"The unique fingerprint characteristics of each chemical compound brought identification of gaseous pollutants possible through comparing the shape, position and relative peak height of each measured spectrum with reference spectra."*. The sentence "any gaseous compounds absorbed in the IR region (approximately 2.5–25 microns) were potential candidates for monitoring using FTIR technology" was moved to **P5L28-29** for a better explanation of the

analytical techniques. All characters in the x and y axes have been enlarged and all 16 species have been included in Figure 4.

**2.) How the rolling background is calculated and used should be explained more detailed.**

Reply: Thanks for the comment. The rolling background was collected using the first spectrum as a background to create an absorbance spectrum from the second spectrum, using the second spectrum as a background for the third spectrum and so on. The integral values of concentrations are calculated to obtain time-series data for each compound. The advantage of using the rolling background is that it will have the best correction for water vapor, detector and instrument response, and the lowest residual error.

**Q5: 2. Materials and Methodology - 2.3 Qualitative receptor modelling:**
**1.) It is not very clear described. What is the index in X_?, X_1, X_2 in Eq 1-4. Or the first index in the coefficient a11. I would assume the index X_1= X_t1 and describe the time of the observations, as the origin of the Factors F1,…is not stated it seems to be taken from the OP-observations. I would like something like F1_op (open path 1) and maybe an corresponding Factor F1_CY o F1_NS , F1_KS. So it is very clear. And please report the factors at the receptor site and the source sites in a comparable way. Maybe complete the factors in table 3 as they are in table 2 just by adding 0.0. And please add the Factors from the site NS to table 3.**

Reply: Thanks for the comment. In order to explain the meaning of each index in **Eq 1-4**

$$X_1 = a_{11}f_1 + a_{12}f_2 + \ldots + a_{1m}f_m + e_1 \qquad \text{Eq. (1)}$$
$$X_2 = a_{21}f_1 + a_{22}f_2 + \ldots + a_{2m}f_m + e_2 \qquad \text{Eq. (2)}$$
$$X_p = a_{p1}f_1 + a_{p2}f_2 + \ldots + a_{pm}f_m + e_p \qquad \text{Eq. (3)}$$
$$\mathbf{X} = (X_1,\ldots,X_P)', \mathbf{f} = (f_1,\ldots f_m)', \text{ and } \mathbf{e} = (e_1,\ldots e_p)' \qquad \text{Eq. (4)}$$

The following sentences have been added to **P6L20-22** -- "where $X_i$ = the i*th* chemical species with mean 0 and unit variance, i = 1,…,p; $a_{i1}$ to $a_{im}$ = the factor loadings for the i*th* chemical species; $f_1$ to $f_m$ = **m** uncorrelated common factors, each with mean 0 and unit variance; e = the error terms indicating the residual part of $X_i$ that is not in common with the other variables". In **Eq 1-4**, "$a_{11}$" represents the factor loading of factor 1 ($f1$) for the first chemical species, "$a_{12}$" represents the factor loading of factor 2 ($f2$) for the first chemical species; "$a_{21}$ represents the factor loading of factor 1 ($f1$) for the second chemical species. $X_1$ describes the communality or common variance of the first chemical species; whereas, $X_2$ represents the communality or common variance of the second chemical species. The factor # based on OP-FTIR (e.g. F1_OP) and the factor # based on CC-FTIR with its corresponding source name (e.g. F1_CY, F1_NS, F1_KS) have been added to Table 2, Table 4, Figure 6 and Figure 7; the factors at the receptor site and the source sites are now presented in a comparable way. The factors from the NS stacks has been added to **table 4 (originally labeled as table 3)** and the following

sentences were added to **P10L33-P11L2** – "The chemicals from the NS stacks were mainly inorganic materials (nitrous oxide, silane, ammonia, nitrous acid, and nitrogen dioxide) that were commonly used in the solar cell production (Table 4e), all of which did not correspond with the organic odorous solvents identified in the receptor sites".

**2.) Maybe you could also report a table with the scalar-products of <F1_CY , F2_OP> for the 4 open path factors Fi_OP with all source factors. Just taking Table 2 and table 3 (after adding the NS factors) would be a 4x (2+2+factors NS) Table/Matrix. It would even be interesting, if the factors of the different sources NS,CY,KS are more or less orthogonal or have a strong overlap.**

Reply: Thanks for the comment. A scalar product is a scalar value that is the result of an operation of two vectors with the same number of components. Given two vectors A and B each with $n$ components, the scalar product is calculated as:

$$A \cdot B = A_1 B_1 + ... + A_n B_n \qquad \text{.Eq.A1}$$

The scalar-products of 4 open path factors (table 2) with all source factors [table 4 (originally labeled table 3)] were calculated using Eq. A1. To perform the calculation of scalar –products, the factor loadings (in table 2 & 4) were replaced by the eigenvectors generated by the SAS programs. The outcomes of a 4 x 6 table/matrix were shown in table A below. It is suggested that none of the scalar-products was orthogonal except for the one calculated from F4_OP · F2_NS, indicating that F4_OP was irrelevant with F2_NS, meaning that F4_OP --"solvent use for paint remover" was irrelevant with F2_NS--"$HNO_3$ thermal decomposition". The top 3 scalar-products were calculated from F1_OP · F1_CY, F1_OP · F1_KS, and F4_OP · F2_KS with scalar-product values of 0.963, 0.727 and 0.570, respectively; indicating that the angles between these three pairs of vectors were far less than 90 degrees, meaning that the relationships between these three pairs of factors were strongest following the order of F1_OP · F1_CY > F1_OP · F1_KS > F4_OP · F2_KS in comparison with the rest of the pairs in this 4 x 6 table/matrix. Therefore, it would suggest that F1_OP--"paint thinner for surface coating" was highly related to F1_CY--"plastic paint thinner"; whereas F1_OP—"paint thinner for surface coating" was closely related to F1_KS--"metal paint thinner"; F4_OP --"solvent use for paint remover" was also related to F2_KS--"cleaner or others". The results of the scalar-products demonstrated that the factors at both the receptor site and the source sites were inter-comparable, which were consistent with the findings in the previous sections of this manuscript.

Table A: The scalar-products of 4 open path factors with all source factors (a 4 x 6 matrix)

| F1_OP · FI_CY | F1_OP · F2_CY | F1_OP · FI_KS | F1_OP · F2_KS | F1_OP · FI_NS | F1_OP · F2_NS |
|---|---|---|---|---|---|
| **0.963** | -0.130 | **0.727** | **0.303** | 0.003 | -0.002 |
| F2_OP · FI_CY | F2_OP · F2_CY | F2_OP · FI_KS | F2_OP · F2_KS | F2_OP · FI_NS | F2_OP · F2_NS |
| -0.072 | 0.001 | -0.076 | 0.005 | -0.179 | 0.074 |
| F3_OP · FI_CY | F3_OP · F2_CY | F3_OP · FI_KS | F3_OP · F2_KS | F3_OP · FI_NS | F3_OP · F2_NS |
| 0.026 | -0.026 | 0.040 | -0.044 | **-0.309** | **0.344** |
| F4_OP · FI_CY | F4_OP · F2_CY | F4_OP · FI_KS | F4_OP · F2_KS | F4_OP · FI_NS | F4_OP · F2_NS |
| -0.092 | **0.349** | **-0.310** | **0.570** | 0.029 | 0.000 |

Note: (1) If A and B are orthogonal (at 90 degrees to each other), the result of the scalar product will be zero; (2) If the angle between A and B are less than 90 degrees, the scalar product will be positive (greater than zero); (3) If the angle between A and B are greater than 90 degrees, the scalar product will be negative (less than zero)

**Q6: 3. Results and Discussion - 3.2 Ambient data from receptor path:**

**1.) Please add a Figure with the time-series which are the basis for the calculation of r_phi and r_pb, OP-FTIR measurements and indicate, when the odor was reported. Figure 2: shows no values, when the factor might be negative, please correct it, even if the contribution of the factor is negative it has to be reported. There should be errors in the coefficients, which explain negative values as least in a small range. Caption Figure 2: Time-series pattern -> Diurnal time-series pattern. Missing Figure: Could you add a complete time series of the 4 factors found at the receptor site, which is not a diurnal pattern.**

Reply: Thanks for the comment. The time-series pattern of chemical species (used as the basis for the calculation of r_phi and r_pb) detected at the receptor site by the OP-FTIR has been added in Figure 5; the yellow highlights in the figure indicated the periods when the odor was reported. The following sentences were added to **P8L27-29** – "*A complete time-series pattern of chemical species found at the receptor site that were used as the basis for the calculation of r_phi and r_pb was shown in Fig. 2, in which the periods when the odor was reported were highlighted*". The negative values in Figure 6a to 6d (originally labeled as Figure 2a to 2d) have been added to the diurnal time-series trends. Caption Figure 6 (originally labeled as Figure 2): "Time-series pattern" has been revised as "Diurnal time-series pattern". A complete time series pattern (not diurnal pattern) of the four factors found at the receptor site has been added in Figure 6e, which suggested that the proportion of the factor scores in negative values were in a relatively small range.

**Q7: 4. Conclusions: Is a bit short and very arbitrary and a little redundant:**

**1.) p.10 line.17: I would replace the "dual-optical sensing system" by "FTIR- spectroscopic measurements." And clarify less ambivalent how the meterological data and cluster analysis was used. Maybe something similar as: "This study developed an alternative investigative framework for detecting air pollution sources of odor nuisance by measuring**

**17 gas species simultaneously using FTIR spectroscopic measurements and factor analyses to identify and characterize emission sources of multiple air contaminants. Meteorological data and Cluster analyses were employed to proof the identification of the major odor emissions" Maybe you could add some numbers how often the odor occurs which originate from CY,KS,NS and the different processes.**

Reply: Thanks for the comment. "Dual-optical sensing system" has been replaced by "FTIR-spectroscopic measurements" in the conclusion section. The role of meteorological data and cluster analysis were clarified by using the sentences suggested by referee 1. The overall content of the conclusion has been rewritten as follows: "*This study developed an alternative investigative framework for detecting air pollution sources of odor nuisance by measuring 16 gas species simultaneously using FTIR spectroscopic measurements and factor analyses to identify and characterize emission sources of multiple air contaminants. Meteorological data and cluster analysis were employed to proof the identification of the major odor emissions. Different industrial processes were related to a specific combination of different pollutants, and this combination was obtained using the two statistical methods of factor analysis and cluster analyses. Factor and cluster analyses were employed to improve the quality and completeness of the source profiles. A field study used FTIR spectroscopic measurements to determine the source of the emission of volatile organic odor species near an industrial park in southern Taiwan demonstrated the feasibility of this proposed method. The major odor emission source was identified through qualitative source apportionment of factor and cluster analyses. With enhanced efficiency in odor investigation methodology, future emission reduction plans can be developed and overall air quality can be improved*".

**Q8: Figures and Table:**
**1.) Please do not use Appendix if you refer to the figure or table in the main text and ensure that all figures and tables are mentioned in the text and keep the order how they are used in the text.**

Reply: Thanks for the comment. The "10. Appendix" section has been removed from the manuscript. All tables and figures originally in "10. Appendix section" was rearranged to either "8. Table" or "9. Figure" sections, in which they were reordered according to how they are used in the text to ensure that all figures and tables are mentioned in the text.

*REFEREE #2:*

The authors present the results of a field study carried out over 10 days in March, 2015, whose goals were to identify and characterize odours reported by commuters at an intersection in an industrial park in southern Taiwan. Most likely candidates were assumed to be industrial VOC emissions from a nearby sunglass factory ("CY") and a metal casings factory ("KS"), which emit organic solvents used in plastic and metal surface coating. Additionally, a nearby solar cell factory ("NS") is a source of inorganic materials used in high-temperature glass sintering.

The experimental approach involved measuring the stack emissions directly with a multi-path closed-cell ("CC") connected to an FTIR spectrometer (effectively a point measurement), as well as indirectly at the intersection with an open-path FTIR system (effectively a path-average measurement over 143 m, one way between FTIR spectrometer and retroreflector array). FTIR measurements at all sites were made at 1 cm$^{-1}$ resolution, averaging 64 IR scans over 5 minutes, and yielding 2,911 consecutive spectra in the case of the open-path system running over 10 days. It is not clear whether the closed-cell measurements at the three factories were done at the same time or separately, but there is 1 day of measurements from CY, 10 days from KS and 4 days from NS. Wind speed and direction measurements were also made near the open-path measurement.

Spectral data analysis to derive VOC species concentrations used multicomponent classical least squares with "rolling backgrounds" in the open-path spectra (subject to changing meteorological conditions) and a "fixed reference method" in the closed-cell spectra (which are sampling stack emissions directly).

Subsequent data analysis consisted of performing a factor analysis involving 4 factors (surface coating, incomplete car engine combustion, solar cell production, solvent use) based on the chemical species observed by open-path FTIR at the receptor site (~16, but there are inconsistencies in the text about which species are being measured and analyzed). The 4 factors (eigenvalues >1) were used in factor loading calculations; chemical species with loadings > 0.4 were considered as influential variables. Factor scores were presented as a function time, separating the one weekend from the remaining weekdays. Additionally, factor scores were presented as windroses.

Species detected at the stacks were compared to ambient open-path data, as well as a database of vehicle exhaust emissions. Finally, factor and cluster analysis were performed on measurements inside the stacks, yielding 2 factors with eigenvalues > 1. One of the factors was identical between stack "CY" and the ambient OP-FTIR data Factor 1, leading the authors to conclude that this was the major odour source.

The study is well-motivated, the experimental setup and data set are valuable, and the data is presented relatively clearly (e.g., dominant factors in bold/red in tables), however, the paper cannot be published without addressing the comments below:

***Q1:*** **First, the paper suffers from a number of distracting inconsistencies between what is described in the text and what is found in the tables and figures. For example, chemical species are referred to in the text that is not in the figures or tables (e.g., cyclohexane, methanol, sulfur hexafluoride, isopropanol, dichloromethane) and the number of species is given as 17 when it is actually 16. Another example involves the discussion of wind directions in Figure 3 regarding OP-FTIR factors, where the named winds do not correspond to the windroses. Another example is a text reference to between 24 and 72 hours of measurements at the stacks but then it states that 2907 spectra were obtained from the KS stack at 5-minute intervals, i.e., a full 10 days of observations.**

Reply: Thanks for the comment. Five chemical species (cyclohexane, methanol, sulfur hexafluoride, isopropanol, dichloromethane) did not meet the following criteria of applying factor analysis in the pilot testing:

(a) Factor loadings under 0.4  (e.g., cyclohexane, factor loading = 0.34),

(b) Kaiser's measure of sampling adequacy (MSA) under 0.5 (e.g., sulfur hexafluoride, MSA= 0.44; Dichloromethane, MSA=0.38), and/or

(c) Final communality estimates (FCE) under 0.5 (e.g., isopropanol, FCE = 0.36).

The text related to the five chemicals should be excluded from the manuscript, but they were overlooked during the proofreading processes. The original locations of these chemicals were as follows: cyclohexane (on P6L27, P8L7, P8L11, and Table A1), Methanol & sulfur hexafluoride (on P7L8), isopropanol & dichloromethane (P7L13). Moreover, the actual number of species was 16, not 17 (corrected on **P7L24 & P8L7**). The inconsistencies of the discussion of wind directions in Figure 7 (original Figure3) were rephrased on **P9L3-5** - "*the incoming direction of these seven species (as represented by factor scores) revealed that the highest factor score occurred in the direction of the WNW, although a few came from the directions of ESE and the directions of NNW-ENE*"**, P9L11-13** –"*The incoming directions of Factor 2 were mostly from NNE–NE, although a few came from the directions of ENE–SE and the direction of NNW (Fig. 7b), indicating multiple source directions for the incomplete engine combustion*"**, P9L14-17** – "*These mainly inorganic compounds exhibited higher concentrations from 06:00 to 09:00 on weekends (Fig. 6c) mostly came from the NNE–ESE directions, although a few came from the SSW direction (Fig. 7c), indicating that the major upwind location of the emission source(s) was located in the NNE–ESE direction*"**, and P9L23** – "*The incoming direction of these two compounds was mainly from the N–ENE direction*". Meanwhile, the source directions described in Table 2 were also revised by the highlighted characters to make it coherent with the directions shown in Figure 7 (original Figure3). The actual hours of measurements at the three stacks were 24 to 242 hours (corrected on **P5L18**).

***Q2:*** **Second, the study claims to develop an "alternative approach" and to demonstrate the feasibility of an "alternative investigative framework", however, a clear description of the details of the methodology, with references to the literature, is insufficient for an**

**uninitiated reader to make use of this "alternative approach". This applies to 1) the spectral analysis, 2) the factor and cluster analysis of the resultant chemical species concentrations, and 3) the grouping of the data as a function of time and wind direction. The paper is quite short so expansion is warranted. Figure and table captions are quite sparse, and tables, in particular, make improper use of footnote numbering to give information without the number applying to anything in particular in the table (see Table 1 and 2).**

Reply: Thanks for the comment. More descriptions of the details of the methodology have been added to section 2.3 & 2.4 described as below:

- **P5L32 to P6L2**: "*The unique fingerprint characteristics of each chemical compound brought identification of gaseous pollutants possible through comparing the shape, position and relative peak height of each measured spectrum with reference spectra.*"

- **P6L4-8**: "*The rolling background was collected using the first spectrum as a background to create an absorbance spectrum from the second spectrum, using the second spectrum as a background for the third spectrum and so on. The integral values of concentrations are calculated to obtain time-series data for each compound. The advantage of using the rolling background is that it will have the best correction for water vapor, detector and instrument response, and the lowest residual error.*"

- **P6L9-12**: "*The fixed reference method uses a reference spectrum that is taken from the zero air or highly purified nitrogen to generate a bundle of spectra using an identical reference spectrum. The main advantage of this method is that the reference is pure, without any contaminants, and the absolute concentrations of the contaminants can be calculated accordingly.*"

- **P7L4-9**: "*Cluster analysis is used to find patterns in a data set by grouping all variables into clusters. A single linkage method (also called nearest neighbor method), a type of hierarchical methods, was used to calculate the distance between two clusters in this study. In the single linkage method, the distance between two clusters A and B is defined as the minimum distance between a point in A and a point in B described as Eq.(5) (Rencher, 2002):*

$$D(A, B) = min\{d(y_i, y_j), for\ y_i\ in\ A\ and\ y_j\ in\ B\} \qquad \text{Eq. (5)}$$

*where d(yi ,yj) is the Euclidean distance*

- **P7L9-14**: "*The concurrent trends between different species can be analyzed using both factor and cluster analysis. Odor contaminants with concurrent patterns were grouped as a factor to gain insight into the underlying emission source characteristics. Meteorological data was used to confirm the factor analysis in the way that the incoming wind direction of each factor (representing a group of chemicals) may be different according to the relative locations of each potential*

*odor sources. Cluster dendrograms provide linkage paths between groups of chemicals to offer more information about the characteristics of different emission sources."*

The captions of tables and figures have been revised in a self-explanatory way:

- Table 1: Descriptive statistics of VOC measurements at the receptor site and the correlation coefficients between the receptor site and the reported odor nuisance events.
- Table 2: The grouping of the data as a function of time and wind direction using factor analysis for chemical species measured by OP-FTIR at the receptor site.
- Figure 1: Trend of total odor nuisance complaints by the TEPA from 2004 to 2017. An increase in odor nuisance complaints has been evidenced in recent years and the odor nuisances have been ranked as the leading cause of environmental nuisances in Taiwan.
- Figure 2: Top view of OP- and CC-FTIR configuration; (a) Receptor path of OP-FTIR monitoring at the intersection. The OP-FTIR beam path was 143 m long in one direction and was equipped with a light emitter on the ground level on one side and a retroreflector at a height of 10 m on the other side. A meteorological station at a height of 12 m was used together with the OP-FTIR beam path to monitor wind speed and direction. Wind and OP-FTIR data were measured as a synchronic system to enable identification of the incoming direction of gaseous contaminants and provide the spatiotemporal measurement of VOCs or odor pollutants; (b) Source stack CC-FTIR measurement at three potential odor emission sources. A 10-m (path length) gas cell with the inner pressure of 720 mm Hg, and an estimated gas flow rate of 0.37 Liter/Sec. was used for the CC-FTIR multi-reflection gas measurements.

The footnotes in Tables 1 and 2 have been revised by using a superscript letter attached to a specific object in the table to give information about that object.

*Q3:* **Third, there are a number of issues with the spectral data analysis. The detection limits in Table 1 are mostly sub-ppb, which seems high for the OP-FTIR technique in general (see, e.g., Jarvis, 2003, "Open Path Spectrophotometry" in the Instrument Engineer's Handbook), more so given the relatively low number of co-added spectra in a 5 minute period, and the relatively high absolute humidity levels (not given in the paper but inferred). The description of the detection limit calculations in Table 1 footnote is not clear. Moreover, it is not clear why CO and $N_2O$ cannot be retrieved from these FTIR spectra. The issue of "rolling backgrounds" is mentioned, but CO and $N_2O$ are far from regions contaminated by water and have been retrieved routinely by others using OP-FTIR, though not with CLS (e.g., Paton-Walsh et al., 2014; Akagi et al., You et al., 2017). Smith et al (2011, AMT) have shown that classical least squares analysis yields inaccurate results at high concentrations (c.f. high concentrations in Table A1 in this study). The nitrous oxide shown in this study in Figure 4, panel (e), has a concentration below 20 ppb (7.9 in Table A1), which is impossible given the**

**ambient concentration of ~330 ppb. Is the accuracy of other retrieved gases similarly affected?**

Reply: Thanks for the comment.
- The MDC in Table 1 represented the estimated minimum detectable concentrations of the OP-FTIR instrument by compound. The values listed in Table 1 should be considered approximations, as the MDC is highly variable, and depends on many factors including atmospheric conditions. MDC can be done based upon the NEA (Noise Equivalent Absorbance) to get a noise limited detection representing the noise level below which no measurement can be made. In this study, the NEA was calculated by a commercial spectral analytical software called "Omnic 7.2a". Both peak-to-peak and RMS noise values can be converted to concentration for a given compound using the peak absorbance shown for that compound in its reference spectrum looking in the analytical region used for analysis. MDC can then be determined by the following equation ("IMACC Open-Path FTIR" the Instrument Engineer's Handbook):

$$\text{MDC} = \frac{(ppm * m)}{A_n(v)} * \frac{NEAx}{pathlength(m)}$$

where

*MDC* = minimum detectable concentration (ppm or ppb),

$A_n(v)$ = normalized absorbance, and

NEAx = noise equivalent absorbance is either the RMS or peak-to-peak noise, giving the RMS or peak-to-peak MDC

Figure A below gives an example of how to calculate the MDC in this study, and it shows that the results of using RMS or peak-to-peak to calculate the MDC were different, the one using RMS noise values yields a lower MDC; whereas, the one using peak-to-peak noise values yields a higher MDC. As the higher MDC is calculated by using the peak-to-peak noise values, we replaced the detection limits (originally calculated by RMS noise value in Table 1) by applying the peak-to-peak noise values to recalculate the detection limits (see Table 1 for revision). The calculated results are intercomparable with the study conducted by the USEPA (e.g., USEPA, 2007, Evaluation of Fugitive Emissions Using Ground-Based Optical Remote Sensing Technology, Table 1-2) based on a longer path length (USEPA =100 m vs. this study = 286m roundtrip) and longer period of co-added spectra (USEPA = 1min vs. this study = 5 min) in our study.

[Figure]

Figure A: An example of calculating Minimum Detectable Concentrations (MDCs)

- The description of the equation of calculating detection limits in Table 1 footnote has been revised to make it clearer to understand by adding the following sentences – "*MDC (estimated minimum detectable concentrations) is calculated by the peak-to-peak (p-p) absorbance noise in the spectral region of the target absorption feature and the MDC is the absorbance signal (of the target compound) that is equal to the p-p noise level, using a reference spectrum acquired for a known concentration of the target compound*".

- We did retrieve CO and $N_2O$ from the field FTIR spectra in this study. The evidence is shown in Figure B below. It demonstrated the comparison between measured spectra (at the receptor site--intersection) and reference spectra (from the spectra library) for both CO and $N_2O$ by using the OP-FTIR. Instead of showing the concentration values in Table 1, the word "detected" was used for CO and $N_2O$, owing to the exact concentration of background species unable to be quantified using a rolling background in the spectral analysis because of the unknown background levels. However, the incremental concentration of these species was still calculated to generate concentration trends suitable for factor analysis.

[Figure]

[Figure]

Figure B: Comparison between measured spectra (at the receptor site) and reference spectra (from the spectra library) for CO and $N_2O$

- The study of Lamp et al. (1997) referred by Smith et al (2011, AMT) used a 20m White cell to measure a broad concentration range of $CH_4$ (8– 1900 ppmm) and CO (10– 3120 ppmm – parts per million meters). The study revealed that the retrieved values were within 5% of the true concentration when the concentrations of $CH_4$ were below 700ppmm; however, accuracy halved at higher concentrations. Similarly, CO retrievals were also suffered from the same type of underestimation when the concentrations were higher than 1000ppmm. This study suggested that classical least squares analysis may yield inaccurate results at high concentrations. However, in our study, Table 3 (originally labeled as Table A1) showed that the concentrations of species detected in the stacks of CY, KS, and NS by a 10m cell ranged from <0.01ppm to 15ppm (or <0.1ppmm to 150ppmm), which were much below the cut points of underestimation for $CH_4$-700 ppmm CO-1000ppmm, according to the results of Lamp et.al. This indicates that the measured concentration in our study may not be affected by the inaccurate result at high concentrations when using classical least square analysis for concentration calculations.

- Because the rolling background method was used to perform the spectral analysis for the OP-FTIR spectra, the ambient concentrations of CO and $N_2O$ (as background species) were not quantified due to unknown background levels. However, the incremental concentration and the integral values of CO and $N_2O$ were still calculated. The concentration of $N_2O$ in Panel (e) in Figure 4 and Table A1 was the incremental concentration that represented the concentrations above the background levels. This means that if the ambient concentration is assumed to be ~330 ppb, the actual concentration of $N_2O$ is 330 (background level) + 7.9 (incremental level) =337.9 ppb. As long as other retrieved gases are not contained in the background environment (just like CO and $N_2O$), their accuracy should not be affected by the interference of background level.

***Q4:*** **Fourth, the authors concede that the first factor might not be limited to one source (P8L18), especially when confronted with Figure 3a, which shows high factor scores from NNE, ESE, and also WNW (last direction not discussed though highest factor score from**

**here). Why the sharp features at WNW and ESE as compared to the more spread out feature at NNW, N, NNE, NE? Also, on P8L30 it is stated that six species coexist at both the CY and KS stacks, but only acetone is truly common, the other 5 are emitted 1000x more at CY – can that be used to separate them?**

Reply: Thanks for the comment.

● According to the OP-FTIR path configuration (as Figure C below), CY was the one located at the WNW location; whereas, KS was the one located at the ESE location; the origin of the high factor scores from WNW and ESE was corresponded to the directions of both CY and KS, indicating the possibility of having more than one source for the first factor. The highest factor score occurred in the direction of WNW was corresponded with the direction of CY.

[Figure]

Figure C: Configuration of OP-FTIR beam path among the three factories (CY, KS, NS)

● According to the windrose and frequency table in Figure D below, the prevailing wind during the 10 days of field monitoring was mainly from NNW (6.63%), N (12.06%), NNE (13.84%), and NE (10.44%). This could lead to the incoming direction of the first factor (representing the seven species) more spread out among NNW, N, NNE, NE in Figure 7a (originally labeled as Figure 3a). The proportion of wind from WNW contributed only 2.68%, in which one half (1.34%) was contributed by low wind speed (0.1-2 ms); the proportion of wind from ESE contributed 7.08%, in which 6.06% was contributed by low wind speed (0.1-2 ms). As CY was the one located at the WNW location; KS was the one located at the ESE location; the sharp features at WNW and ESE were probably related to the fact that the emission sources were located at these two directions. The concentration of the first factor chemicals may increase when the wind turned to WNW or ESE directions, although the proportion of wind from these two directions was relatively low compared to others. Lower wind speed at WNW or ESE was another trigger resulting in an increase of concentrations because the odor contaminants (the first factor chemicals) were trapped in the area adjacent to the emission sources when the wind is calm.

[Figure]

0309-0319_Windrose

| Wind direction | Calm (<0.1 ms) | Wind speed (m/s) 0.1~2 | 2~3 | 3~4 | 4~5 | 5~6 | >6 | Total(calm excluded) |
|---|---|---|---|---|---|---|---|---|
| N | 0.07% | 6.15% | 3.09% | 1.55% | 0.69% | 0.38% | 0.21% | 12.06% |
| NNE | 0.14% | 9.41% | 3.40% | 0.72% | 0.21% | 0.10% | 0.00% | 13.84% |
| NE | 0.10% | 7.94% | 1.92% | 0.41% | 0.10% | 0.07% | 0.00% | 10.44% |
| ENE | 0.14% | 4.40% | 0.96% | 0.17% | 0.03% | 0.00% | 0.00% | 5.57% |
| E | 0.07% | 4.16% | 0.45% | 0.14% | 0.00% | 0.03% | 0.03% | 4.81% |
| ESE | 0.03% | 6.05% | 0.89% | 0.07% | 0.07% | 0.00% | 0.00% | 7.08% |
| SE | 0.03% | 4.47% | 0.96% | 0.14% | 0.03% | 0.00% | 0.00% | 5.60% |
| SSE | 0.14% | 2.95% | 1.06% | 0.45% | 0.14% | 0.03% | 0.00% | 4.64% |
| S | 0.03% | 2.75% | 1.82% | 1.13% | 0.55% | 0.21% | 0.00% | 6.46% |
| SSW | 0.00% | 1.89% | 1.41% | 0.72% | 0.31% | 0.00% | 0.03% | 4.36% |
| SW | 0.03% | 1.58% | 1.03% | 0.34% | 0.03% | 0.03% | 0.00% | 3.02% |
| WSW | 0.00% | 2.16% | 1.48% | 0.38% | 0.07% | 0.00% | 0.00% | 4.09% |
| W | 0.00% | 2.03% | 1.51% | 0.65% | 0.07% | 0.03% | 0.00% | 4.29% |
| WNW | 0.03% | 1.34% | 0.89% | 0.38% | 0.03% | 0.03% | 0.00% | 2.68% |
| NW | 0.07% | 2.23% | 0.65% | 0.34% | 0.14% | 0.07% | 0.00% | 3.44% |
| NNW | 0.10% | 3.71% | 1.58% | 0.65% | 0.55% | 0.10% | 0.03% | 6.63% |
| SUM | 1.00% | 63.21% | 23.12% | 8.24% | 3.02% | 1.10% | 0.31% | 99.01% |

Figure D: Integrated windrose graph and the frequency table of wind data

(3) Among the species found in the CY and KS stacks, six species (ethyl acetate, toluene, o-xylene, m-xylene, p-xylene, and acetone) coexisted in both factories. The relatively higher concentration of acetone (compared to other 5 species at the KS stacks) might not be sufficient to separate the six species commonly existed at both CY and KS stacks because the chemicals used in both CY and KS were organic solvents that are similar to each other.

*Q5:* **Fifth, the authors name CY as the source of the odours based on an identical Factor 1 composition as compared to OP-FTIR. Does this check out with a windrose plot of butyl cellosolve and PGMEA, which are unique to CY based on Table A1? For that matter, can it be verified that when winds blow from stack NS (nearly from the East, not NE as given on P7L34 and P8L1) cyclohexane, acetylene and ethylene increase? There should also be toluene and xylene. I can understand why no correlation plots (like Figure 5) are shown for KS given that the correlation is discussed as below 0.1, but why is NS not discussed in this way at all?**

Reply: Thanks for the comment.
- Two unique compounds - butyl cellosolve and PGMEA were found only in the CY stacks. Figure E shows the radar plots of butyl cellosolve and PGMEA, which also confirmed that these two unique odorous compounds came from the direction of CY (WNW-NNW).

[Figure]

Butyl cellosolve

[Figure]

PGMEA

[Figure]

Figure E: Radar plots for butyl cellosolve and PGMEA

- The direction of the stack NS (nearly from the East, not NE) were rewritten as **P9L18**– "*The solar cell production company located in the East direction and using inorganic materials such as ammonia, silane, and nitric acid to produce silicon glass….*".

- Figure F (d, e, f) demonstrated radar plots of cyclohexane, acetylene, and ethylene, indicating that their incoming directions were not restricted to a specific direction. The concentrations of acetylene and ethylene did increase when the wind blows from stack NS, however, the increases also evidenced when the wind blows from stack CY. Based on Table 3 (originally labeled as Table A1), ammonia, nitrous oxide, and nitrogen dioxide were the unique compounds found in the NS stacks (CC-FTIR), which also appeared in the receptor path (OP-FTIR). Radar plots of ammonia and nitrogen dioxide shown in Figure F (b, c) indicated that their concentration did increase specifically when the wind blows from stack NS on the East (and ENE). Although ammonia was also found in KS stacks, the concentration levels of NS stacks were 28 times higher than that of KS stacks, showing that NS stacks contributed more ammonia than the KS stacks. Moreover, the radar plots of toluene and xylene (Figure G) indicated that the major source direction for toluene and xylene was more related to the direction of CY (WNW).

[Figure]

Figure F: Radar plots of ammonia, nitrogen dioxide, cyclohexane, acetylene, and ethylene

[Figure]

Figure G: Radar plots for toluene and xylene

- Table 4e showed that the chemicals from the NS stacks were mainly inorganic materials (nitrous oxide, silane, ammonia, nitrous acid, and nitrogen dioxide) that did not correspond with the organic odorous solvents identified in the receptor sites. The reason why only CY has its correlation plots delineated in the manuscript is because the discussion about the major emission source has come to the last paragraph of the whole manuscript; at this point, we have to conclude where the major odorous emission source (CY) is and to narrow down the discussion before the conclusion can be made. Nevertheless, the correlation scatter plots of selected compounds for NS stacks that were also found in receptor path (except for silane) are delineated as the Figure H below, some of them even exhibited negative correlation coefficients (e.g., $NH_3$ vs. $N_2O$). At the receptor path, the correlation coefficient was mostly below 0.2, except the relatively higher correlation coefficient of $NH_3$ vs. $NO_2$ (r= 0.46). One possible reason for this inconsistency is that not all stacks in the NS plants were measured by the CC-FTIR (4 out of 10 stacks were measured), owing to some technical concerns raised by the plant operators. On the other hand, all 7 stacks in CY were monitored at once during the period of monitoring.

[Figure]

[Figure]

| Stack NS (CC-FTIR) | Receptor Path (OP-FTIR) |

Figure H: Scatter plots of concentration variations over time between two selected contaminants from NS stacks (CC-FTIR) and receptor path (OP-FTIR)

***Q6:*** **Sixth, why is Factor 2 (incomplete engine combustion) clustered to the NW and SE (Figure 3b, not discussed), whereas the source directions are indicated as "all directions" in Table 2? It also appears like there are roads all around the open-path in Figure 1. Are some roads more major than others?**

Reply: Thanks for the comment. The source directions indicated as "all directions" in Table 2 have been revised as NNE-NE, ENE-SE, NNW. The following paragraphs have been added to the content of Factor 2 to explain more clearly about the source directions (including NW and SE) in **P9L11-13**. The roads surrounding the OP-FTIR path were almost equal in size.

***Q7:*** **Seventh, the time-series of factor scores do not correspond to factory working hours in an obvious way. Do the plants run 24 hours a day? Does the solar cell plant run on weekends? Why is there a traffic peak from 6-8 on the weekend? Why does the traffic factor look similar to the solvent factor?**

Reply: Thanks for the comment. Yes, most plants in this industrial park run 24 hours a day and 7 days a week. The solar cell plant runs on weekends as well; the official operational permit showed that the operation hours of the solar cell plant were 24 hours a day, and 360 days a year. The peak from 6-8 on the weekends may be contributed by traffic on the weekends (working 7 days a week).The negative values in Figure 6a to 6d (originally labeled as Figure 2a to 2d) have been added to the diurnal time-series trends (as requested by referee 1). A timeseries pattern of factor scores of the four factors was also added to Figure 6e (as requested by referee 1), which indicated that the original factor scores of factor 2 and factor 4 were not as similar as those shown in the diurnal time-series pattern (Figure 6b and 6d). Except for the weekends (3/14-3/15), the pattern of incomplete engine combustion (factor 2) revealed more or less a regularly "twice a day" pattern during the weekdays. However, the pattern of solvent use (factor 4) revealed a type of continuous trend, which is different from the type of intermittent trend shown in factor 2. The different pattern of factor score for factor 2 and factor 4 can also be compared as Figure I below (identical to Figure 6e).

[Figure]

Figure I: The original time-series trends of factor 2 (incomplete engine combustion) and factor 4 (solvent use)